# Higher-order singularities in phase-tracked electromechanical oscillators

Xin Zhou [1,8] ✉, Xingjing Ren[1,8], Dingbang Xiao[1], Jianqi Zhang[2], Ran Huang[3], Zhipeng Li[4], Xiaopeng Sun[1], Xuezhong Wu[1] ✉, Cheng-Wei Qiu [4], Franco Nori [3,5] ✉ & Hui Jing [6,7] ✉

Singularities ubiquitously exist in different fields and play a pivotal role in probing the fundamental laws of physics and developing highly sensitive sensors. Nevertheless, achieving higher-order (≥3) singularities, which exhibit superior performance, typically necessitates meticulous tuning of multiple (≥3) coupled degrees of freedom or additional introduction of nonlinear potential energies. Here we propose theoretically and confirm using mechanics experiments, the existence of an unexplored cusp singularity in the phase-tracked (PhT) steady states of a pair of coherently coupled mechanical modes without the need for multiple (≥3) coupled modes or nonlinear potential energies. By manipulating the PhT singularities in an electrostatically tunable micromechanical system, we demonstrate an enhanced cubic-root response to frequency perturbations. This study introduces a new phase-tracking method for studying interacting systems and sheds new light on building and engineering advanced singular devices with simple and well-controllable elements, with potential applications in precision metrology, portable nonreciprocal devices, and on-chip mechanical computing.

Singularities, sometimes referred to as catastrophes, arise in diverse disciplines and play an essential role in describing how the properties of an object, that are dependent on certain controlling parameters, change qualitatively even if the controlling parameters vary minimally[1,2]. The unusual landscapes near these singularities are very useful for enhancing the sensitivities of detection[3–12], suppressing noise[13–17], as well as generating nonreciprocity[8,9,18–30]. Higher-order singularities have the potential to provide higher performance and engender richer physics[10–17,31–35]. However, constructing and adjusting such higher-order singularities is typically challenging due to the requirement for multiple (≥3) coupled degrees of freedom[10,31–33].

Interestingly, nonlinearities can facilitate the emergence of higher-order singularities, such as dynamical "pitchfork" bifurcation points[12–17,36–43] and higher-order exceptional points (EP's)[11,34,35], while requiring fewer degrees of freedom. Exploring these phenomena not only expands our understanding of singularity dynamics but also paves the way for engineering more controllable singular devices. Nevertheless, these nonlinearities are often associated with well-established nonlinear potential energies.

The study of novel singularities in optical systems has been conducted extensively[8,9]. However, thus far, the exploration of novel singularities in micro/nanoelectromechanical systems, which exhibit

[1]College of Intelligence Science and Technology, NUDT, 410073 Changsha, China. [2]State Key Laboratory of Magnetic Resonance and Atomic and Molecular Physics, Wuhan Institute of Physics and Mathematics, Innovation Academy of Precision Measurement Science and Technology, Chinese Academy of Sciences, 430071 Wuhan, China. [3]Center for Quantum Computing, Cluster for Pioneering Research, RIKEN, Wako-shi, Saitama 351-0198, Japan. [4]Department of Electrical and Computer Engineering, National University of Singapore, Singapore 117576, Singapore. [5]Department of Physics, University of Michigan, Ann Arbor, MI 48109-1040, USA. [6]Key Laboratory of Low-Dimensional Quantum Structures and Quantum Control of Ministry of Education, Department of Physics and Synergetic Innovation Center for Quantum Effects and Applications, Hunan Normal University, 410081 Changsha, China. [7]Academy for Quantum Science and Technology, Zhengzhou University of Light Industry, 450002 Zhengzhou, China. [8]These authors contributed equally: Xin Zhou, Xingjing Ren. ✉e-mail: zhouxin11@nudt.edu.cn; xzwu@nudt.edu.cn; fnori@riken.jp; jinghui73@gmail.com

broad applications[44–51], exceptional in-situ controllability[42,43,52–56], and rich interactive phenomena[52–58], remains relatively limited.

Here, we demonstrate theoretically and experimentally the existence of an unexplored third-order singularity in the phase-tracked (PhT) steady states of a pair of coherently coupled mechanical modes. Notably, by examining the equiphase contour of the coherent-coupling phase response, we find that the system can exhibit bistability in a way qualitatively different from the Duffing nonlinearity. The boundaries of stability are constituted by a series of saddle-node bifurcation points, leading to the singularity named folds[1,37] to describe the abrupt transitions that occur during parametric sweeping across these boundaries. Two folds tangentially merge at a "pitchfork" bifurcation point referred to as a nexus[31], which defines a cusp singularity if projected onto the parameter plane[1,37]. By investigating the state information associated with these singularities, we find that these correspond to transitions between oscillation phases characterized by chirality. Experimental validation of the cusp singularity is achieved in an electrically tunable microelectromechanical resonator. Our findings demonstrate that the PhT cusp singularity enables enhanced detecting sensitivity, exhibiting a cubic-root response that surpasses binary EP singularities.

## Results
### Concept

We investigate a pair of coherently coupled mechanical modes characterized by adjustable natural frequencies $\omega_{1,2}$ and matched dissipation rate $\gamma$, as conceptually depicted in Fig. 1a. In this study, the coherent coupling is produced by the rotation-induced Coriolis effect[59], presenting an angular velocity $\Omega$-dependent coupling strength $g = 2\kappa\Omega$, where $\kappa \approx 0.85$ represents the Coriolis-coupling coefficient. One of the modes, namely mode 1, experiences linear excitation through an applied external sinusoidal force denoted as $F_0 \cos(\omega_d t)$, while the second mode, mode 2, is not driven. The linear displacement response of each mode is mathematically described as $q_{1,2} = |q_{1,2}| \cos(\omega_d t + \theta_{1,2})$, wherein $|q_{1,2}|$ and $\theta_{1,2}$ correspond to the amplitude and phase responses, respectively.

In the scenario where the two modes reach degeneracy ($\Delta\omega \equiv \omega_2 - \omega_1 = 0$), the open-loop amplitude-frequency response $|q_1|$ of the driven mode exhibits normal mode splitting as a function of the coupling strength $g$[52]. Correspondingly, the associated phase response $\theta_1$ of the driven mode is visualized by the colored surface in Fig. 1b. Here, we analyze the "tomography" of the driven-mode phase response, by keeping $\theta_1$ a constant oscillation phase $-\pi/2$. To achieve this PhT closed-loop oscillation, a phase-locked-loop (PLL) is

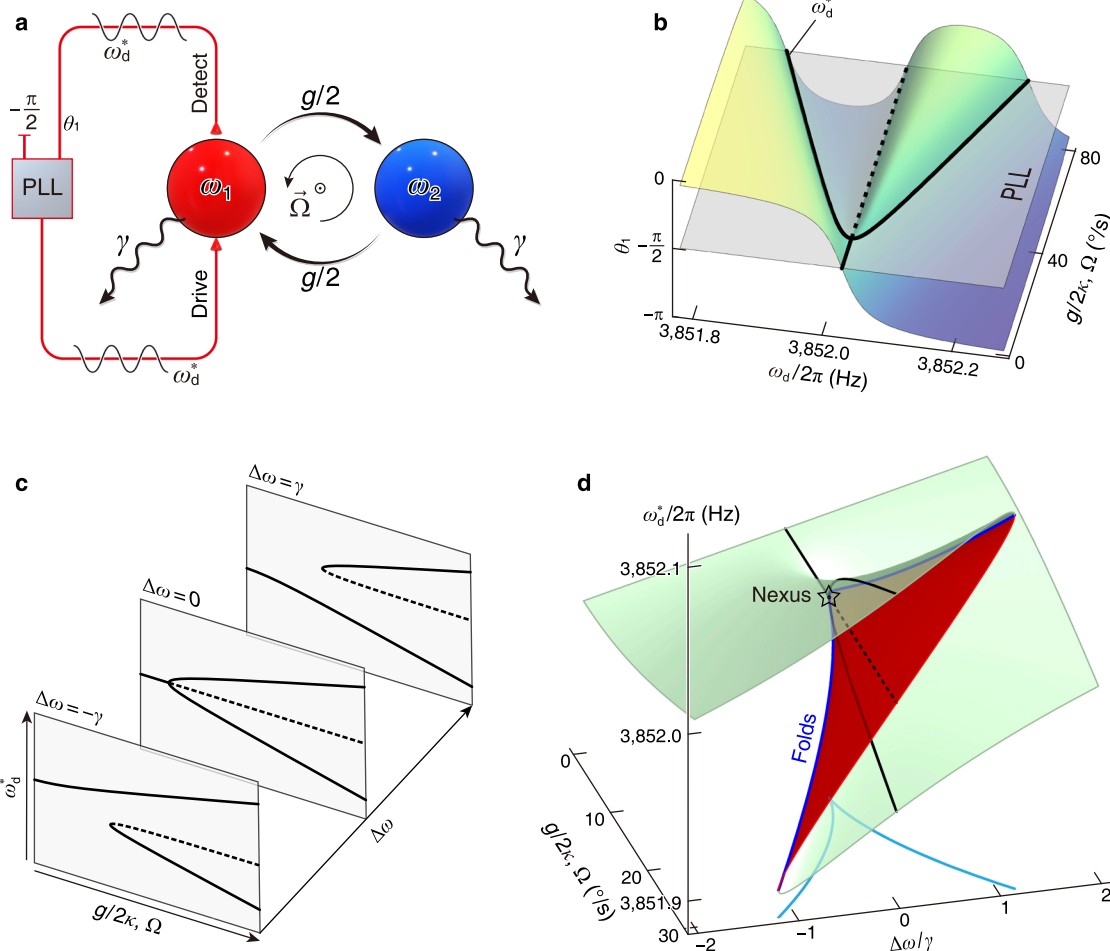

**Fig. 1 | Higher-order singularities in phase-tracked (PhT) dynamics: concept. a** Schematic representation of the realization. Mode 1, driven by an external force, is coherently coupled to mode 2, which remains free. A PLL is employed to enable PhT closed-loop oscillations. In this study, the coherent coupling is produced by the rotation $\vec{\Omega}$ induced Coriolis effect, resulting in a coupling strength of $g = 2\kappa\Omega$, with $\kappa = 0.85$. **b** Open-loop phase-frequency response ($\theta_1$, colored surface) of the driven mode 1 as a function of the coupling strength $g$ in the degenerate case, $\Delta\omega = 0$. The

PLL adjusts the drive frequency to track the phase $\theta_1 = -\pi/2$. The PhT frequency $\omega_d^*$ exhibits a "pitchfork" bifurcation. **c** Bifurcation patterns of $\omega_d^*$ at typical degeneracy conditions. **d** The $\omega_d^*$ as a function of degeneracy condition $\Delta\omega$ and coupling strength $g$. The green (red) region of the surface represents the stable (unstable) regime. The projection of the stability boundaries (blue curves) made up of the bifurcation points to the parameter plane manifests two parabolic loci merged at a cusp (cyan curves).

implemented, as depicted in Fig. 1a. By examining the black contour in Fig. 1b, we observe that the PhT closed-loop frequency (referred to as $\omega_d^*$) satisfying the condition $\theta_1 = -\pi/2$ exhibits a "pitchfork" bifurcation relative to the coupling strength $g$. Notably, this bifurcation arises solely from the landscape of the linear phase response $\theta_1$, distinguishing it from its counterparts that relay on nonlinear potential energies[12,37,42]. Remarkably, this "pitchfork" bifurcation point is precisely located at the threshold between weak and strong coupling.

As the degeneracy is broken, the perturbed "pitchfork" bifurcation of $\omega_d^*$ splits into a saddle-node bifurcation and a stable branch, as illustrated in Fig. 1c. Through the continuous adjustment of $\Delta\omega$, $\omega_d^*$ manifests as a partially folded 3D surface (Fig. 1d). The functional relationship between the PhT frequency $\omega_d^*$, the coupling strength $g$, and the degeneracy condition $\Delta\omega$ can be accurately described by the cubic equation (see Supplementary Note 3):

$$(\omega_d^* - \omega_1)\left(\omega_d^* - \omega_2 + \frac{i}{2}\gamma\right)\left(\omega_d^* - \omega_2 - \frac{i}{2}\gamma\right) - \frac{1}{4}g^2(\omega_d^* - \omega_2) = 0, \quad (1)$$

which describes a cusp singularity because equ. (1) is right-equivalent to the universal unfolding of Thom's codimension-two catastrophe[60,61] (see Methods).

The inflectional region (highlighted in red) within the folded $\omega_d^*$ surface in Fig. 1d is made by the unstable bifurcation branches. This instability arises due to the system's pronounced divergence when subjected to perturbations (see Supplementary Note 4). If the control parameters $g$ and $\Delta\omega$ steer across the stability boundaries made by the saddle-node bifurcation points adiabatically, catastrophic jumps in the oscillation state take place, defining the singularity called folds. The folds tangentially merge at the "pitchfork" bifurcation point (star in Fig. 1d), giving rise to a nexus and a markedly twisted $\omega_d^*$ geometry. The projection of the nexus onto the $\Delta\omega$-$g$ parameter plane defines a cusp singularity[1,37]. The singularities of folds and cusp are mathematically characterized by the discriminant of the cubic Eq. (1) (details see Supplementary Note 5).

## Experimental realization

To implement the configuration illustrated in Fig. 2a, we utilize a pair of four-node standing-wave modes in a capacitive microelectromechanical disk resonator[62]. These modes denoted as 1 and 2, possess nearly degenerate natural frequencies $\omega_{1,2}/2\pi \approx 3.85$ kHz, alongside equivalent dissipation rates $\gamma = 2\pi \times 55.8$ mHz. Notably, the deformations of these modes strictly adhere to an in-plane pattern. In Fig. 2b, we present a micrograph portraying an identical device to the one employed in our experimental setup. Our device is the core of a high-performance micro gyroscope[48,49,62] (see Supplementary Note 1 for more details).

The coherent coupling between modes 1 and 2 is achieved through the rotation-induced Coriolis effect (see Supplementary Note 2). In Fig. 2a, we indicate the distributions of vibrational velocity for the modes along the outline of the disk resonator. The red, blue, and magenta arrows represent the radial, tangential, and total velocities of different mass points, respectively. The resonator is mounted on a rotating rate table, a specialized device in the inertial system industry that can provide precise rotational movement, to facilitate the out-of-plane rotation at a controlled angular velocity $\Omega$. The Coriolis force acting on each mass point is determined by the cross product of the rotation vector $\vec{\Omega}$ and the velocity vector. The Coriolis force distribution caused by the radial (tangential) velocity distribution of one mode is proportional to the tangential (radial) velocity distribution of the other. Collectively, the rotation introduces vibrational interactions between the near-degenerate modes, characterized by a strength denoted as $2\kappa\Omega$. When transformed from the standing-wave to the traveling-wave basis, the Coriolis coupling can be interpreted as a

rotational Doppler effect[63], also regarded as an acoustic analog of the Zeeman effect[18].

The experimental setup is shown in Fig. 2b (see Supplementary Note 1 for more details.). To drive mode 1 into linear vibration, two alternating actuation signals are selectively applied on the electrodes positioned at the antinodes of mode 1. The differential driving configuration effectively eliminates the undesired crosstalk actuation to mode 2. The antinodal displacements of the two modes, $q_{1,2}$, are transduced through charge amplifiers and subsequently detected using a lock-in amplifier based on the Homodyne method (see Methods). By applying a direct current tuning voltage $V_t$ to the electrodes located at the antinodes of mode 2, we are able to modify the natural frequencies $\omega_{1,2}$ (Fig. 2c), thereby facilitating adjustments to the degeneracy condition $\Delta\omega$ through the introduction of electrostatic negative stiffness (see Methods). The $\theta_1 = -\pi/2$ PhT oscillations can be realized by enabling the PLL in Fig. 2b.

We commence our analysis by examining the open-loop frequency responses of the system when the PLL is deactivated. In Fig. 2d, e, and f, we present the amplitude ($|q_{1,2}|$) and phase ($\theta_{1,2}$) responses as functions of the angular velocity ($\Omega$) and driving frequency ($\omega_d$) under different degeneracy conditions of $\Delta\omega = 0, -\gamma$, and $\gamma$, respectively. In the degenerate case where $\Delta\omega = 0$, the system enters the strong-coupling region, when the Coriolis-coupling rate surpasses the dissipation, as expressed by $2\kappa\Omega \geq \gamma$. The presence of normal mode splitting, evident in the $|q_1|$ responses shown in Fig. 2d, leads to mode hybridization. The eigenfrequencies are precisely determined by $\omega_\pm = [\omega_1 + \omega_2 \pm (\Delta\omega^2 + 4\kappa^2\Omega^2)^{1/2}]/2$ (dot-dashed curves in the $|q_1|$ responses). Notably, the Coriolis coupling induces vibrations in mode 2, as evidenced by the $|q_2|$ responses. In the $\theta_1$ responses, the $\theta_1 = -\pi/2$ equiphase contours accurately reproduce the bifurcation patterns predicted in Fig. 1c. The "pitchfork" bifurcation point is located at $\Omega_0 = \gamma/(2\kappa)$. In cases where the degeneracy is broken ($\Delta\omega \neq 0$), the symmetry of normal mode splitting in the $|q_1|$ responses is broken, and the $\theta_1 = -\pi/2$ equiphase contour in the $\theta_1$ responses illustrates a stable branch and a saddle-node bifurcation (Fig. 2e, f).

To investigate the PhT states, we activate the PLL, which serves to regulate the driving frequency $\omega_d$, in order to maintain the phase at the set value $\theta_1 = -\pi/2$. We first adjust the value of $V_t$ to ensure $\Delta\omega \approx 0$, and vary $\Omega$ adiabatically, ranging from zero to 80°/s. The PhT frequencies obtained from both experimental measurements and theoretical calculations are presented in Fig. 2g. When the angular velocity falls below the strong-coupling threshold, $\omega_d^*$ remains locked to $\omega_1$. However, at the threshold point (cusp), $\Omega_0 = 11.83$°/s, $\omega_d^*$ transitions randomly to one of the two stable bifurcation branches. In Fig. 2g, the upper stable branch is experimentally observed.

Next, we proceed to modify $\Delta\omega$ by adjusting $V_t$ adiabatically while maintaining $\Omega$ at specific predetermined values. The variation in the PLL-controlled $\omega_d^*$ is presented in Fig. 2h. The experimental results are depicted by the blue dashed (upward) and red solid (downward) curves, showcasing the outcomes of $V_t$ sweeps in opposite directions. If $\Omega > \Omega_0$, the sweeping curves encounter abrupt discontinuities at certain $V_t$ values known as catastrophes or singularities. A hysteresis loop is formed by the two upward and downward curves at the same $\Omega$, with its size decreasing as $\Omega$ is reduced until it disappears when $\Omega \leq \Omega_0$. The observed singularities, mapped to the $V_t$–$\Omega$ parameter plane, are shown in Fig. 2i, which coincide well with our theoretical predictions (see Supplementary Note 5).

## State information

In the following, we will delve into the details of the state information corresponding to each PhT frequency. We will show that the "pitchfork" bifurcation is caused by the breaking of chiral symmetry, and the singularities are associated with transitions of oscillation phases with different chiralities.

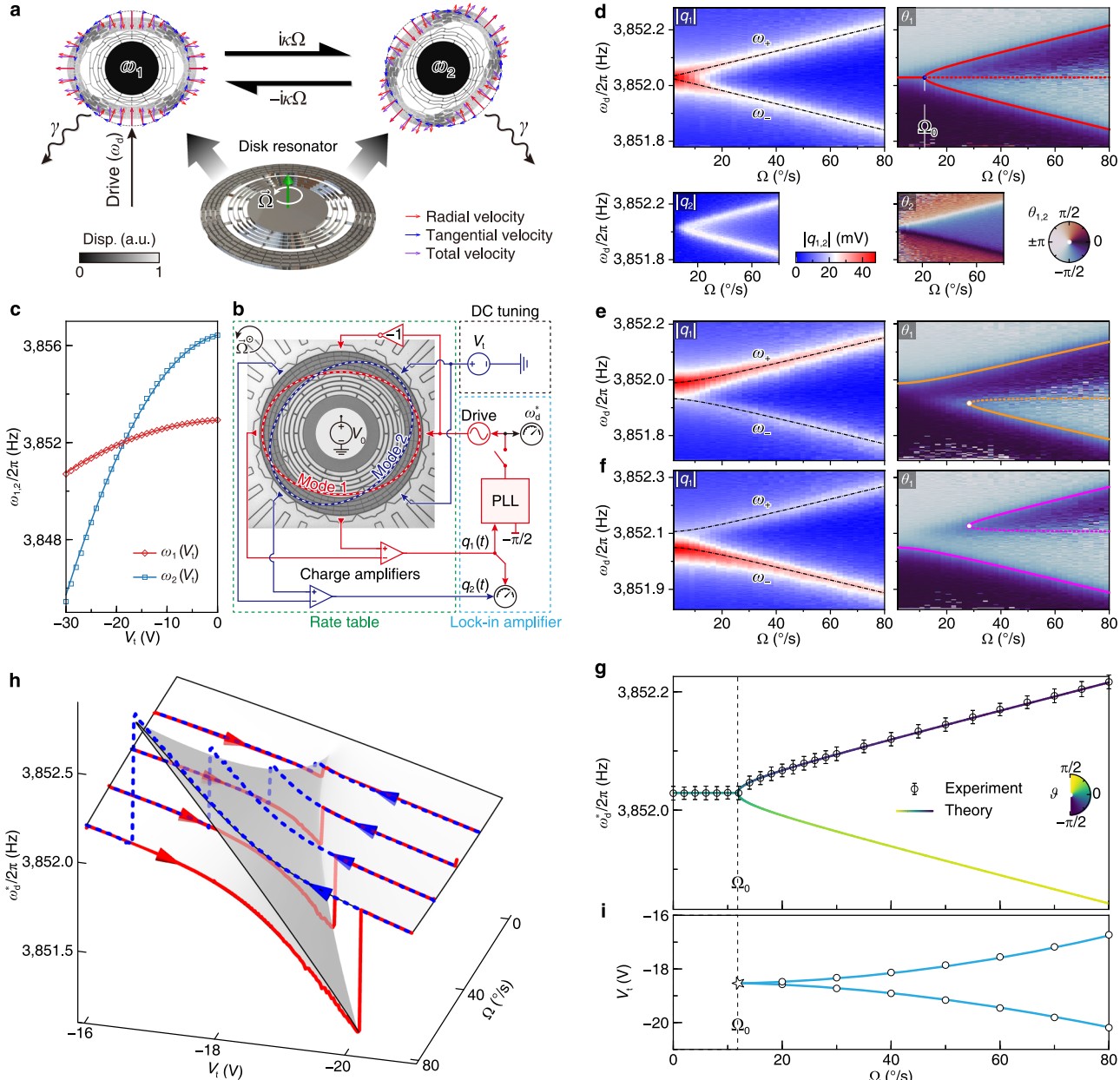

**Fig. 2 | Experimental realization of higher-order singularities in micro-electromechanics. a** Two near-degenerate in-plane standing-wave modes of a microelectromechanical disk resonator are used to realize the scheme in Fig. 1a. **b** Experimental setup. Mode 1 is driven differentially by a force $F_0 \cos(\omega_d t)$, while mode 2 remains unexcited. The device is mounted on a rotating rate table to introduce an out-of-plane rotation $\vec{\Omega}$. The charge amplifiers transduce the anti-nodal displacements of both modes, $q_{1,2}$, which are then recorded and demodulated by a lock-in amplifier, to yield their amplitude and phase responses. The PhT condition is enabled by activating the PLL to lock the phase of mode 1 in quadrature, $\theta_1 = -\pi/2$. The degeneracy condition $\Delta\omega$ can be adjusted by applying an electrostatic tuning voltage $V_t$ to the antinodal electrodes of mode 2. **c** Natural frequencies of the modes, $\omega_{1,2}$, versus tuning voltage $V_t$. **d–f** Experimental

frequency responses of the amplitude and phase versus the angular velocity $\Omega$ for the degeneracy conditions $\Delta\omega \approx 0$ (**d**), $-\gamma$ (**e**), and $\gamma$ (**f**). The dot-dashed curves in the $|q_1|$ responses represent the eigenfrequencies. Colored contours in the $\theta_1$ responses indicate the $\theta_1 = -\pi/2$ PhT frequency $\omega_d^*$, confirming (**d**) the "pitchfork" bifurcation and (**e**, **f**) the saddle-node bifurcations. **g** PhT frequency $\omega_d^*$ measured by the PLL versus the angular velocity $\Omega$ at degeneracy. The error bars are the standard deviation. The colored curve is the theoretical result. **h** PLL measured $\omega_d^*$ when the tuning voltage $V_t$ is adiabatically swept at constant angular velocities. The blue dashed (red solid) curves depict the $V_t$-increasing (decreasing) sweeps, illustrating singularities and hysteresis if $\Omega_0 > \Omega_0$. The gray surface is the theoretical result. **i** Singularities projected onto the $V_t$-$\Omega$ plane. The white-faced points (light blue curves) are experimental (theoretical) data.

The PhT state can be described by the vector $|\psi\rangle = \cos\frac{\phi}{2}|1\rangle + e^{i\vartheta}\sin\frac{\phi}{2}|2\rangle$, where $\{|1\rangle, |2\rangle\}$ represents the orthonormal basis of modes 1 and 2, $\phi \equiv 2\arctan(|q_2|/|q_1|)$ represents the polar angle, and $\vartheta = \theta_2 - \theta_1$ represents the relative phase or azimuthal angle. We emphasize that all states involved in this study are classical. As shown in Fig. 3a, this state vector can be projected onto a classical Bloch sphere with coordinates $(S_1, S_2, S_3)^T$, where $S_1 = \sin\phi\cos\vartheta$,

$S_2 = \sin\phi\sin\vartheta$, and $S_3 = \cos\phi$ stand for the ellipticity, chirality, and orientation, respectively (see Supplementary Note 6). Each state is represented by a polarization pattern within the $q_1$–$q_2$ plane. The regions of instability, bistability, and monostability on the Bloch sphere are depicted in light red, light green, and gray colors, respectively. The front and back hemispheres correspond to PhT states with positive and negative angular velocities, respectively.

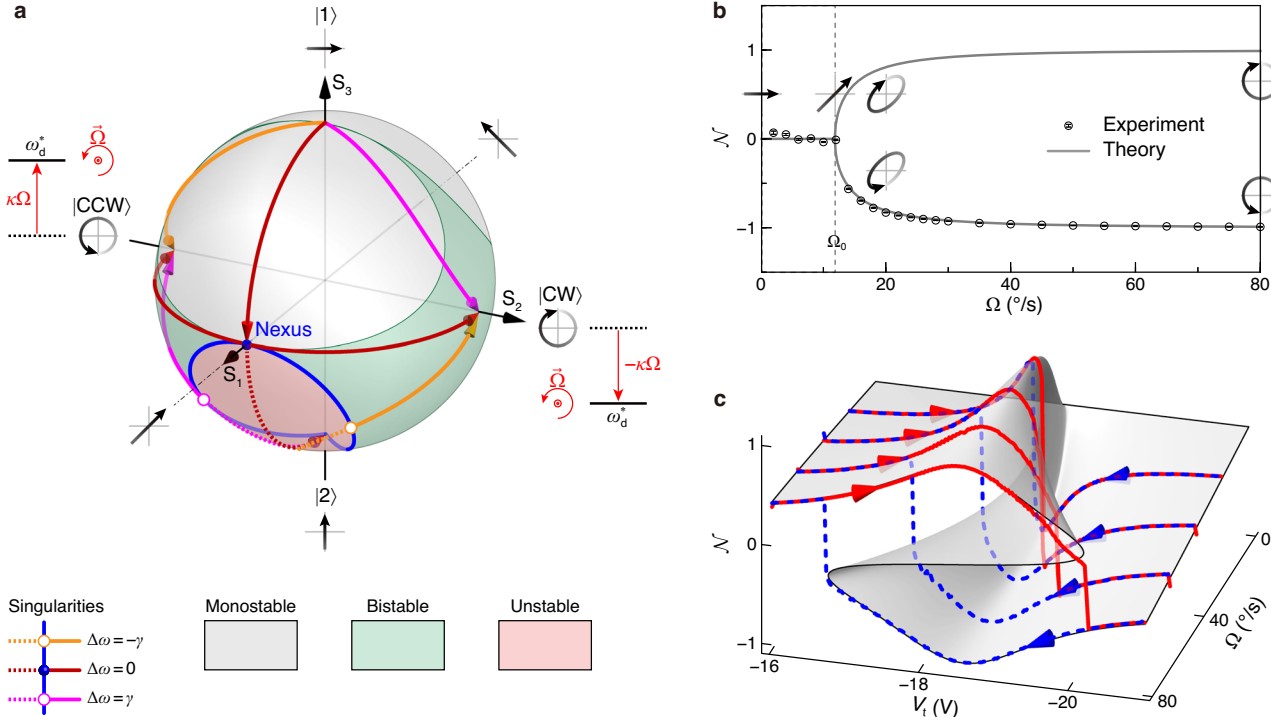

**Fig. 3 | State information. a** Classical Bloch sphere describing the PhT states. The red, orange, and magenta trajectories represent the state evolutions corresponding to the bifurcation patterns for degeneracy conditions $\Delta\omega = 0, -\gamma$, and $\gamma$, respectively. The arrows indicate the $\Omega$-increasing direction. The blue curve represents the singularities, which are composed of a series of bifurcation points. The PhT frequency of the $|CCW\rangle$ ($|CW\rangle$) increases (decreases) because of the rotational Doppler effect. **b** The order parameter $\mathcal{N}$ corresponding to the "pitchfork" bifurcation measurement in Fig. 2g, illustrating the spontaneous breaking of chiral symmetry at $\Omega_0$, or a second-order transition of oscillation phase. Here, $\mathcal{N}$ is defined as the chirality. Error bars are the standard deviation. **c** $\mathcal{N}$ corresponds to the catastrophe measurement in Fig. 2h. Singularities or catastrophes can be considered as first-order transitions of different oscillation phases. The gray surface is the theoretical result.

The state evolution of the "pitchfork" bifurcation when $\Delta\omega = 0$ is shown by the red trajectories on the Bloch sphere in Fig. 3a. Initially, at $\Omega = 0$, the system is initialized in state $|1\rangle$, characterized by horizontally linear polarization. As $\Omega$ is increased to reach the weak–strong-coupling threshold $\Omega_0$, the system evolves into a state exhibiting 45° linear polarization (blue point), where $|q_1| = |q_2|$ and $\vartheta = 0$. Upon further increase of $\Omega$ beyond $\Omega_0$, the state bifurcates into three distinct branches. The middle branch is unstable, leading the system to randomly transition to one of the degenerate stable branches. These stable branches correspond to states predominantly exhibiting either clockwise circular polarization, represented as $|CW\rangle = (|1\rangle + i|2\rangle)/\sqrt{2}$, or counter-clockwise circular polarization, represented as $|CCW\rangle = (|1\rangle - i|2\rangle)/\sqrt{2}$. This process signifies a spontaneous breaking of chiral symmetry. The PhT frequency $\omega_d^*$ of the $|CW\rangle$ ($|CCW\rangle$) dominant state increases (decreases) due to the Doppler effect induced by the positive rotation[18,63], resulting in the frequency bifurcation.

In the degeneracy-broken cases ($\Delta\omega \neq 0$), the state evolutions associated with the bifurcation patterns in Fig. 2e, f are shown by the orange and magenta trajectories on the Bloch sphere in Fig. 3a, respectively. The introduction of rotation immediately leads to the breaking of chiral symmetry, as shown by the stable branches on the upper hemisphere. A series of saddle-node bifurcation points (white-faced points) on the lower hemisphere constitute the folds (blue curves). Two folds tangentially merge at the "pitchfork" bifurcation point referred to as the nexus (blue point), forming a cusp singularity.

Subsequently, we demonstrate that the frequency singularities or catastrophes are associated with transitions of different oscillation phases. To capture the variations in oscillation phases, we introduce the order parameter $\mathcal{N}$ as the relative population of the chiral states

$|CW\rangle$ and $|CCW\rangle$, thereby characterizing distinct oscillation phases (see Supplementary Note 7). Specifically, the order parameter is defined as $\mathcal{N} \equiv \frac{\langle\psi|CW\rangle\langle CW|\psi\rangle - \langle\psi|CCW\rangle\langle CCW|\psi\rangle}{\langle\psi|CW\rangle\langle CW|\psi\rangle + \langle\psi|CCW\rangle\langle CCW|\psi\rangle} = \sin\phi\sin\vartheta$, which equals the chirality. The process of spontaneous symmetry breaking underlying the "pitchfork" bifurcation shown in Fig. 2g is illustrated in Fig. 3b. This process represents a second-order transition from the chiral symmetric oscillation phase ($\mathcal{N} = 0$) to the chiral-symmetry broken oscillation phase ($\mathcal{N} \neq 0$). The second-order oscillation phase transition point is associated with the cusp singularity.

The order parameters corresponding to the upward (downward) sweeps depicted in Fig. 2h, are shown by the blue dashed (red solid) curves in Fig. 3c. These curves signify first-order transitions from the $|CCW\rangle$ ($|CW\rangle$) dominant oscillation phase to the $|CW\rangle$ ($|CCW\rangle$) dominant oscillation phase. The first-order oscillation phase transition points are associated with the fold singularities.

## Cubic-root sensitivity

It has been revealed that the singularities are very sensitive to parameter perturbations[3–9,12], owing to the sharp changes in topology near these points. Here, we demonstrate that the PhT singularity nexus exhibits an enhanced cubic-root sensitivity to perturbations, surpassing that of the conventional binary EP singularities[3–6].

In Fig. 4a, we observe that at the singularity nexus ($\Omega = \Omega_0$ and $\Delta\omega = 0$), the PhT frequency $\omega_d^*(\Omega_0)$ aligns with the natural frequency of the driven mode, $\omega_1$. Otherwise, if the degeneracy is broken $\Delta\omega \neq 0$, $\omega_d^*(\Omega_0)$ deviates suddenly but continuously from $\omega_1$. This deviation, $\delta\omega_X = \omega_d^*(\Omega_0) - \omega_1$, demonstrates a sharp change when $\Delta\omega$ shifts away from the nexus, as shown in Fig. 4b. To assess the impact of perturbations that can affect the degeneracy condition, denoted as $\epsilon$ ($\sim\Delta\omega$), we consider the sensing output $\delta\omega_X$ of $\epsilon$ in the vicinity of the nexus, as

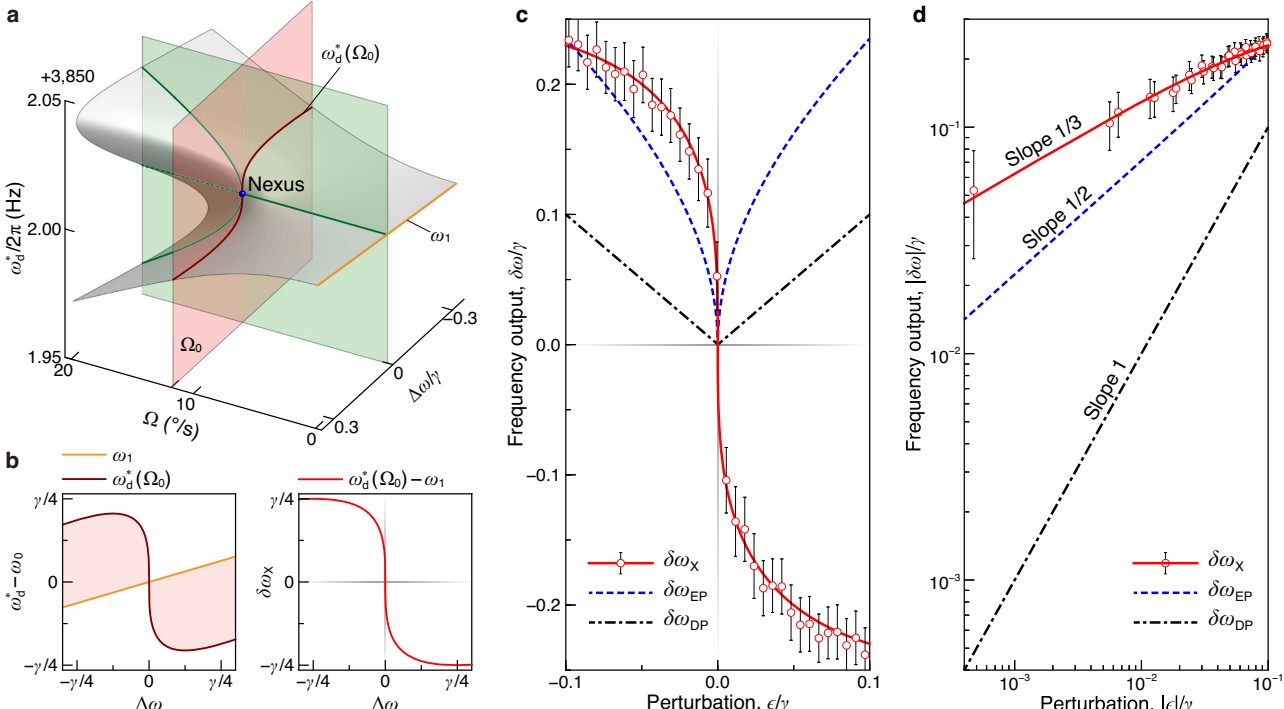

**Fig. 4 | High sensitivity near the PhT cusp singularity. a** The PhT frequency $\omega_d^*$ as a function of angular velocity $\Omega$ and degeneracy condition $\Delta\omega$. The contours of $\Omega = \Omega_0$ (dark red curve) and $\Delta\omega = 0$ (green curves) portray the sharp variation of $\omega_d^*$ near the singularity nexus (blue point). **b** The PhT frequency at the critical angular velocity $\omega_d^*(\Omega_0)$ and its shift from $\omega_1$, $\delta\omega_X = \omega_d^*(\Omega_0) - \omega_1$ as functions of $\Delta\omega$. Here, $\omega_0$ represents $\omega_1$ at $\Delta\omega = 0$. In the range of $-0.25\gamma \leq \Delta\omega \leq 0.25\gamma$, Frequency output $\delta\omega_X$ decreases monotonically with $\Delta\omega$. **c** Frequency output $\delta\omega_X$ near the singularity

nexus versus the natural-frequency perturbation $\epsilon = \Delta\omega$ from both simulation (red solid curve) and experiment (red circles). The eigenfrequency splits near an EP (blue dashed curve) and a DP (black dot-dashed curve) are also simulated. Error bars are the standard deviation. **d** Logarithmic plot of the absolute data in **c**. The PhT cusp singularity has a cubic-root output, providing higher sensitivity compared to the EP and DP.

illustrated by the red curve in Fig. 4c (see Supplementary Note 8). On a logarithmic scale, this sensing output exhibits a cubic-root response near the nexus: $\delta\omega_X \sim \epsilon^{1/3}$, as shown in Fig. 4d, confirming the cubic nature of the singularity nexus. To experimentally verify this cubic-root behavior, we maintain a fixed rotation rate $\Omega_0$ and introduce a fine-tuning voltage $V_t$ to sweep across the nexus. By converting $V_t$ to $\epsilon$ (see Methods), the experimental input-output data are represented by the red circles in Fig. 4c, d, which coincide well with the cubic-root simulation.

We conduct a comparison between the sensitivities generated by the PhT singularity nexus and a binary EP singularity, which is produced by a passive parity-time-symmetric system with a chosen damping difference equal to the dissipation of our system (see Supplementary Note 8). The blue dashed curves in Fig. 4c, d reveal a dependency of $\epsilon^{1/2}$ for the eigenfrequency split $\delta\omega_{EP}$ near the EP. It is noteworthy that the sensitivity exhibited by the PhT singularity nexus exceeds that of the binary EP[3–6], and is on par with the third-order EP[10]. Moreover, both $\delta\omega_{EP}$ and $\delta\omega_X$ demonstrate significant improvements, when compared to the standard output $\delta\omega_{DP} \sim \epsilon$ of the diabolic-point (DP) system, as denoted by the black dot-dashed curves in Fig. 4c, d.

## Discussion

In summary, our study has discovered a cusp singularity in the phase-tracked coherent-coupling dynamics of a pair of microelectromechanical modes. By utilizing highly controllable elements, our finding enables the construction of advanced singularities. This discovery holds promise for engineering novel electromechanical devices and opens up new possibilities for phase-related interactive dynamics investigations in various fields, including optics, optomechanics, and hybrid quantum systems. Furthermore, we present an alternative approach for creating bistability and bifurcations by establishing a

phase-tracked closed-loop oscillation in a coupled system without relying on nonlinear potential energy. This not only enhances our understanding of closed-loop oscillation dynamics but also extends coherent control into the singularity region.

The PhT singularity holds potential for various applications such as precise sensing, rapid mode switching, and mechanical computing[64]. The abstraction of the closed-loop oscillations into bits, independent of vibration amplitudes, offers potential advantages in terms of power consumption and lifetime. Additionally, the PhT cusp catastrophe can also facilitate the realization of closed-loop controlled nonreciprocal state transfer (see Supplementary Discussion 1). In contrast to previous studies that rely on two-parameter-controlled encircling[24–30,34,35], we demonstrate the achievement of nonreciprocal state transfer through the highly desirable single-parameter (voltage)-controlled traversal, resulting in an impressive isolation ratio of 59 decibels. Moreover, the PhT cusp singularity resulting from Coriolis coupling can be directly used to enhance gyroscope sensitivity and achieve deep-sub-linewidth mode matching.

While our experimental demonstration focuses on Coriolis coupling, it is theoretically possible to realize the same effects using ordinary linear coherent coupling (see Supplementary Discussion 2). Future research can delve into PhT singularities originating from different types of coupling[53,55,65–68], explore the interplay between different kinds of singularities, and investigate phase-tracked dynamics in many-body systems with an increased number of degrees of freedom[32,69,70].

## Methods

### Electrostatic frequency tuning

The tuning voltage $V_t$ introduces electrostatic negative stiffness to both modes 1 and 2, given by $\omega_{1,2}^2(V_t) = \omega_{1,2}'^2(0) - T_{1,2}(V_0 - V_t)^2$, where

$\omega'_{1,2}(0)$ represents the natural frequencies of the bare mechanical modes. The electrostatic tuning factors $T_{1,2}$ are proportional to the capacitive area, the inverse modal mass, and the inverse cubic of the capacitive gap. The stiffness perturbation induced by $V_0$ exists even in the absence of the tuning voltage $V_t$ and can be included in the intrinsic natural frequencies. By defining $\omega_{1,2}^2(0) = \omega'^2_{1,2}(0) - T_{1,2}V_0^2$, and assuming that the electrostatic stiffness perturbation is small relative to the intrinsic stiffness, we have

$$\omega_{1,2}(V_t) = \sqrt{\omega_{1,2}^2(0) + T_{1,2}(2V_0 V_t - V_t^2)}$$
$$\approx \omega_{1,2}(0) + K_{1,2}(2V_0 V_t - V_t^2). \tag{2}$$

Here, the tuning coefficients are defined as $K_{1,2} = T_{1,2}/[2\omega_{1,2}(0)]$.

The experimentally measured natural frequencies $\omega_{1,2}$ as functions of $V_t$ are represented by the red and blue data points in Fig. 1c. These data points are fitted (curves) to the model (2) with parameter values $\omega_1(0) = 2\pi \times 3852.92$ Hz, $\omega_2(0) = 2\pi \times 3856.43$ Hz, and $V_0 = 2.5$ V. The fitted tuning coefficients are $K_1 = 1.29 \times 10^{-2}$ rad s$^{-1}$ V$^{-2}$ and $K_2 = 6.40 \times 10^{-2}$ rad s$^{-1}$ V$^{-2}$. Furthermore, the relationship between the difference of natural frequencies (degeneracy condition), $\Delta\omega = \omega_2 - \omega_1$, and the tuning voltage $V_t$ can be expressed as:

$$\Delta\omega(V_t) = \Delta\omega(0) + (K_2 - K_1)(2V_0 V_t - V_t^2), \tag{3}$$

where $\Delta\omega(0) = \omega_2(0) - \omega_1(0)$.

**Homodyne measurement**

The capacitive transducers pick up the antinodal displacements of the two micromechanical modes, represented as $q_1$ and $q_2$, which can be expressed as $q_j = |q_j|\cos(\omega_d t + \theta_j)$ for mode $j$ ($j = 1, 2$). These signals are then converted to voltage signals by the integrated charge amplifiers on a printed circuit board. Finally, a two-channel lock-in amplifier (Zurich Instruments HF2LI) is used to record the voltage signals. To determine the amplitudes $|q_{1,2}|$ and phases $\theta_{1,2}$ relative to the driving signal, dual-phase demodulation techniques are employed. Specifically, the process involves splitting $q_j(\omega_d, t)$ and individually mixing it with the driving reference signal $\cos(\omega_d t)$ and a copy of it that is phase-shifted by $\pi/2$. The equations representing this mixing process are as follows,

$$|q_j|\cos(\omega_d t + \theta_j) \times \cos(\omega_d t) = \frac{|q_j|}{2}\left[\cos\theta_j + \cos(2\omega_d t + \theta_j)\right],$$
$$|q_j|\cos(\omega_d t + \theta_j) \times \cos\left(\omega_d t + \frac{\pi}{2}\right) = \frac{|q_j|}{2}\left[\sin\theta_j - \sin(2\omega_d t + \theta_j)\right].$$

After removing the high-harmonic components using low-pass filters, the in-phase component $X_j = \frac{|q_j|}{2}\cos\theta_j$ and the quadrature component $Y_j = \frac{|q_j|}{2}\sin\theta_j$ are obtained. By transforming these components into polar coordinates, we can derive the amplitude $|q_j|$ and phase $\theta_j$ as follows:

$$|q_j| = \sqrt{X_j^2 + Y_j^2}, \theta_j = \arctan\frac{Y_j}{X_j}.$$

**Codimension-two nature of the PhT singularity**

The governing equation (1) of the PhT singularity can be expanded as

$$\delta^3 - \frac{\Delta\omega}{2}\delta^2 + \frac{1}{4}(\gamma^2 - \Delta\omega^2 - g^2)\delta + \frac{\Delta\omega}{4}(\gamma^2 + \Delta\omega^2 + g^2) = 0, \tag{4}$$

where $\delta \equiv \omega_d^* - (\omega_1 + \omega_2)/2$. The ordinary cubic equation (4) is right-equivalent to

$$X^3 + AX + B = 0, \tag{5}$$

which is the universal unfolding of Thom's codimension-two cusp catastrophe[60,61]. Here, the new variable is defined as $X = \delta - \Delta\omega/6 = \omega_d^* - (\omega_1 + 2\omega_2)/3$, and the two parameters $A$ and $B$ are given by

$$A = \frac{1}{4}\left(\gamma^2 - \frac{4}{3}\Delta\omega^2 - g^2\right),$$
$$B = \frac{\Delta\omega}{4}\left(\frac{2}{3}\gamma^2 + \frac{8}{27}\Delta\omega^2 + \frac{g^2}{3}\right),$$

respectively. In other words, the PhT singularity of this study is classified as a codimension-two cusp singularity. The codimension-two nature indicates that one has to control at least two parameters to construct such a cusp-embedded surface.

## Data availability
Data relevant to the figures and conclusions of this manuscript are available at https://doi.org/10.6084/m9.figshare.19609350.

## Code availability
The codes used for the numerical calculations are available from the corresponding author upon reasonable request.

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

## Acknowledgements

X.Z. thank Prof. Ashwin Seshia from the University of Cambridge, Prof. Zenghui Wang from the University of Electronic Science and Technology of China, and Prof. Chun Zhao from the University of York for helpful discussions. This work is partly supported by the National Natural Science Foundation of China (NSFC) grant U21A20505 (D.X., X.Z., and X.W.), grant 11935006 (H.J.), grant 51905539 (X.Z.), the Hunan Provincial Major Sci-tech Program grant 2023zj1010 (H.J. and X.Z.), the Science and Technology Innovation Program of Hunan Province grant 2020RC4047 (H.J.), the Young Elite Scientist Sponsorship Program by CAST grant YESS20200127 (X.Z.), the Natural Science Foundation of Hunan Province for Excellent Young Scientists grant 2021JJ20049 (X.Z.). F.N. is supported in part by Nippon Telegraph and Telephone Corporation (NTT) Research and the Foundational Questions Institute Fund (FQXi) via Grant No. FQXi-IAF19-06. This work is primarily supported by the National Key R&D Program of China (NKPs) grant 2022YFB3204901 (X.Z.).

## Author contributions

X.Z. conceived the idea and designed the research. X.Z. and X.R. performed the experiments. X.Z., H.J., F.N., C.-W.Q., J.Z., R.H., and Z.L. conducted the theory. X.Z. designed the device. X.Z. and D.X. fabricated the device. X.Z., X.R., X.S., D.X., and X.W. developed the test circuitry. X.Z., H.J., and F.N. wrote the manuscript with inputs from all authors. The project was jointly supervised by X.Z., H.J., and F.N.

## Competing interests

The authors declare no competing interests.
