## [Peer Review File · Nature Communications]

REVIEWER COMMENTS

Reviewer #2 (Remarks to the Author):

Report on "Cubic singularities in binary linear electromechanical oscillators" by Xin Zhou et al.

The submitted manuscript introduces a micromechanical resonator as a platform to realize cubic singularities in a coupled binary system without involving nonlinearities. The idea is novel and could be of interest to specialists in higher-order singularities in binary systems.

I regret, however, that I cannot conclude that the paper provides the sort of clear advance in scientific understanding that would likely excite the interest of Nature Communications' readership. The present manuscript would find a more appropriate audience in a journal that publishes more specialized research.

A significant issue with the manuscript is that it is challenging to follow. Without a careful reading of the Supplementary Information, it is impossible to understand the details of the mechanical resonator, the experimental setup, and the measurements. Moreover, the description of the resonator and the experiment is minimal. I realized that the micromechanical resonator is encapsulated in a vacuum, so an SEM picture of the device is not possible, but a more detailed description of the rationale behind the device's design would be helpful. Why are there nine concentric rings, and why the inner ones are different than the outer rings? Does the ring-down measurement show in Figure 1d start at $t=0$ or $t=2$ seconds? Coherently coupled mechanical modes exhibit constant amplitude oscillations even after turning off the external driving force.

Using the Coriolis force to couple the modes coherently is an interesting idea. The authors should explain this phenomenon in more detail in the main document. Moreover, the experimental setup used to rotate the micromechanical resonator and the read-out circuitry must be described in more detail. There is only a brief mention of it in the Supplementary Information document. How fast the "temperature-controlled high precision rate table" rotates? Is the out-of-plane wobble reported small enough that it won't induce mode coupling? The micromechanical resonator reported is pretty big: a mechanical device with a diameter of around 10 mm won't be considered micro-mechanical nowadays and will exhibit considerable out-of-plane vibrations.

Another attractive result of the manuscript is the demonstration of a physical system where the nonreciprocity can be controlled electrostatically. It would be helpful for the readers to include a discussion on how much the nonreciprocity can be augmented in linear systems using the new "knob."

The manuscript claims that "although we experimentally demonstrate the phase tomographic closed-loop cubic singularity based on the Coriolis coupling, in principle, it can also be realized using ordinary linear coherent coupling." I don't understand this statement. If this is true, what is the advantage of the Coriolis force? A linear coherent coupling could be implemented using electronic circuitry among the resonators, which significantly simplifies the implementation of this idea.

In summary, the manuscript reports an interesting prototype to study singularities in coupled binary systems without involving nonlinearities, but it requires substantial modifications and does not seem a good fit for Nature Communications.

Reviewer #3 (Remarks to the Author):

The paper describes an experimental observation of hysteretic phenomena in a system of two coupled micromechanical modes. The detuning of the modes and the coupling are the control parameters. One of the modes is driven close to resonance. The central (albeit unannounced) idea is that the system can be made effectively nonlinear, and that this is accomplished by fixing the phase difference between the vibrations of the driven mode and the drive. Unexpectedly, this nonlinearity leads to the multi-branch response and, respectively, to hysteresis. The qualitative features of the emerging behavior are familiar from the bifurcation theory, but this theory had not been applied to the system studied in the paper. The observations are new. I find them interesting.

Some details: The coupled modes are standing waves formed by whispering gallery modes in a micromechanical system that has the shape of a disk. The circular symmetry of the system makes the modes almost degenerate. Rotation of the disk leads to mode coupling via the Coriolis force. The coupling is controlled by the angular frequency. One of the modes is driven, its phase is measured, and the displacements of the modes are detected. The phase difference between the vibrations of the driven mode and the drive is set to $\pi/2$.

The frequency of the drive that maintains the phase difference is a function of the mode coupling and the frequency detuning. It satisfies a cubic equation. If plotted as a surface with respect to the control parameters, it has two folds that merge at the nexus. The folds are the sets of the bifurcation values of the control parameters.

A few comments now.

I think the paper has to be re-written. It has real results, there is no need in overselling them. Terms like "phase-tomographic singularity" are irrelevant and confusing.

The “stability” has to be defined. In the text there is a reference to the stability of the drive frequency, but I believe that the stability refers to the mode vibrations. The corresponding equations have to be provided and loss of stability when crossing the folds has to be demonstrated.

The picture of the drive frequency as a function of the parameters is a standard picture of the bifurcation theory, see Ref. 1. It is drawn in multiple textbooks and monographs (a recent one is Kamenev’s “Field Theory of Non-Equilibrium Systems”). There are standard names for the corresponding bifurcations, co-dimension 1 for the folds (usually the saddle-node bifurcation) and co-dimension 2 for the cusp. These terms have been broadly used in micro- and nanomechanics, for example in the analysis of the bistability of the response of nonlinear modes to a resonant drive. The analysis can be and should be formulated in standard terms.

To continue on the terminology: There are no phase transitions in the studied system and there are no phases of matter, there are phases of the modes. Calling hysteresis “nonreciprocity” does not add to the understanding, just makes it confusing to see what is being discussed. Calling the phase shift between the modes “circular polarization” does not help to understand what is going on. Panel (d) in Fig. 2 makes an impression that the “circular polarization phase” emerges immediately after crossing the bifurcation point, which is incorrect, I guess. Besides, the term “phase” here refers, judging from the context, to the “phase of matter”, although this is in fact a vibrational phase.

I believe the mapping on the Bloch sphere is counterproductive. It makes the formulation look “quantum”, but rather distracts from what is going on. The modes are dissipative, and there is no Poisson bracket between the variables. Projecting the folds on the Bloch sphere instead of the standard projection on a plane makes the picture confusing.

If the authors want to put emphasis on the phase shift between the modes and the change of their amplitude, they could use the Stokes parameters. This is unnecessary, but at least this is a standard language for a two-mode system where the modes vibrate at the same frequency.

I note also that the high sensitivity of a mechanical system near the cusp point (nexus) has been demonstrated by Aldridge and Cleland [PRL 94, 156403 (2005)]; the authors demonstrated that, in fact, the sensitivity can be made exponentially strong if one looks at the response to fluctuations in the corresponding parameter range. But the scaling of the dynamics is certainly the same as in the present paper.

Given the novelty of the results, a significantly revised version of the paper can be considered for publication in Nature Communications.

Response to Reviewers

We express our sincere gratitude to the reviewers for their detailed reviews. The manuscript has been modified considerably to address the feedback received. Please find below our response to the reviewers. Reviewers' comments are in **black**, authors' responses are in **Blue**, and the **Red bullets** indicate the corresponding modifications in the manuscript.

Reviewer #2 (Remarks to the Author):

Report on “Cubic singularities in binary linear electromechanical oscillators” by Xin Zhou et al.

The submitted manuscript introduces a micromechanical resonator as a platform to realize cubic singularities in a coupled binary system without involving nonlinearities. The idea is novel and could be of interest to specialists in higher-order singularities in binary systems. I regret, however, that I cannot conclude that the paper provides the sort of clear advance in scientific understanding that would likely excite the interest of Nature Communications' readership. The present manuscript would find a more appropriate audience in a journal that publishes more specialized research.

A significant issue with the manuscript is that it is challenging to follow. Without a careful reading of the Supplementary Information, it is impossible to understand the details of the mechanical resonator, the experimental setup, and the measurements. Moreover, the description of the resonator and the experiment is minimal. I realized that the micromechanical resonator is encapsulated in a vacuum, so an SEM picture of the device is not possible, but a more detailed description of the rationale behind the device's design would be helpful. Why are there nine concentric rings, and why the inner ones are different from the outer rings? Does the ring-down measurement shown in Figure 1d start at $t=0$ or $t=2$ seconds? Coherently coupled mechanical modes exhibit constant amplitude oscillations even after turning off the external driving force.

Response: We would like to express our sincere thanks to the Referee for the valuable feedback, which is very constructive for revising the manuscript. We apologize for the weak description of the device in our initial manuscript. In fact, the device used to demonstrate the singularities in this study is the core of an established MEMS gyroscope called the disk resonator gyroscope (DRG). As one of the most precise MEMS gyroscopes, its bias instability can reach inertial grade, as reported in Ref. [1-5] of the Supplementary Information. The high performance of the DRG is due to its superb controllability, excellent tuneability, good stability, and high signal-to-noise ratio of the displacement transduction. The advantages mentioned above and the integration nature of this device make it a very convenient experimental platform for basic research (e.g. Ref. [6.7] of the Supplementary Information). The singularities reported in this study are not restricted by the structural design of our disk resonator. We just demonstrated the singularity physics in our established device, which was

previously designed in Ref. [1] of the Supplementary Information for the purpose of enhancing the transduction signal-to-noise ratio.

As the Referee suggested, we have made additional introductions of the mechanical resonator, the experimental setup, and the measurements in the main text, to make the main story comprehended completely without referring to the Supplementary Information. Moreover, more detailed descriptions of the device, experimental setup, and measurements are included in Supplementary Note 1.

The device we tested is indeed encapsulated in a ceramic leaded chip carrier, rendering it imperceptible to visual observation. The picture in the manuscript was taken using another device with an identical design, which is not the very device employed in our experiments. The vague representation in the original manuscript has been corrected.

The ring-down signal in our original manuscript might be misleading, we thank the Referee for pointing this out. We have redefined the origin of the measuring time to avoid this misunderstanding in the revised Supplementary Figure 1d.

Here, we would like to provide some clues about the potential implications of this paper for the kind consideration of the Referee. Singularities, as exotic phase transitions emerging naturally in diverse fields, have many surprising features and intriguing functionalities in practical applications (e.g. Ref. [2-15] of the main text). Higher-order (≥ 3) singularities can provide even higher performances (Ref. [16-21] of the main text). However, thus far, the only way to realize and control high-order singularities is to utilize multipartite or highly nonlinear systems, which turns out to be very challenging for many fields in practice. To overcome this difficulty, this paper reports the first realization of a cubic singularity in a binary system, without the need for multi-mode (≥ 3) interactions or even nonlinearity. Four outcomes of our discovery make this manuscript stand out from related studies: (1) Our method may open new avenues for building and engineering advanced singular devices with simple and well-controllable elements, thus stimulating more studies and applications of singularity physics. (2) Our cubic singularity provides a highly practical and efficient method for enhancing sensitivities for gyroscopes, mass spectrometers, electrometers, gravimeters, and other sensors. (3) The compactness of the phase space engenders rich topological physics, our work may open up a new tomographic dimension for studying such interactive physics in phase space. (4) We present an alternative approach for creating cubic bifurcations by establishing a phase-tracked closed-loop oscillation in a coupled system without relying on nonlinearity. This not only enhances our understanding of closed-loop oscillation dynamics but also extends coherent control into the singularity region.

Changes made:

- We have revised the representation of the *Experimental realization* part of the main text: From Line 121 to Line 131, the device is described in more detail. We make a statement that the micrograph is a device identical to the one we test. From Line 132 to Line 151, we describe the modes and the Coriolis coupling in detail. From line 152 to Line 166, we provide adequate information for the experiment setup.

From Line 167 to Line 187, the open-loop characterization is introduced to show the equi-phase contours. From Line 188 to Line 212, the phase-tracked closed-loop characterization that observes the singularities is represented.

- More details about the device, experimental setup, and measurements are included in Figure 2 of the main text. Figure 2a provides information on the device, mode shapes, and the Coriolis coupling. Figure 2b provides the experimental setup. Figure 2d-f provide the open-loop characterization results.
- In Supplementary Note 1, a more detailed description of the device is included in the first paragraph.
- The ring-down result in Supplementary Figure 1d is revised. The measuring time is started when the drive is turned off. The corresponding description in Line 53 of the Supplementary Information is also revised.
- The experimental setup is described in more detail in Section B. Experimental setup of Supplementary Note 1. Supplementary Figure 3 is extended to show more details about the signal-processing circuitry and experimental environment.

Using the Coriolis force to couple the modes coherently is an interesting idea. The authors should explain this phenomenon in more detail in the main document. Moreover, the experimental setup used to rotate the micromechanical resonator and the read-out circuitry must be described in more detail. There is only a brief mention of it in the Supplementary Information document. How fast does the "temperature-controlled high precision rate table" rotate? Is the out-of-plane wobble reported small enough that it won't induce mode coupling? The micromechanical resonator reported is pretty big: a mechanical device with a diameter of around 10 mm won't be considered micro-mechanical nowadays and will exhibit considerable out-of-plane vibrations.

Response: We really thank the Referee for pointing out the important concern about the experiment testbed. As the Referee suggested, the Coriolis coupling effect has been explained in the revised main text. The experimental setup used to rotate the micromechanical resonator and the read-out circuitry have been described in detail in Section B. Experimental setup of Supplementary Note 1 and Supplementary Figure 3.

The rate table is a commonly used characterization equipment in gyroscope tests. The rate table used in this study can provide a stable programmed angular velocity in a range of $0^\circ/\text{s}$ to $\pm 1200^\circ/\text{s}$. The precision of the applied angular velocity at stable rotation can be $< 0.001^\circ/\text{s}$. During the test, the device and the circuitry are fixed on the rate table, and electrically connected to the outside equipment through slip rings, such that the device and circuitry can perpetually rotate without affecting the electric connection with the equipment.

The operational modes in this study are perfectly in-plane, and there is no out-of-plane wobbling in this gyroscopic resonator. We apologize for the vague description of the mode vibration. In the revised manuscript, we have made a clearer description of this.

As mentioned previously, this device is a core of an established MEMS gyroscope. Our previous study indicates that the 8-mm diameter disk resonator can resist more than 1000g of shock acceleration and is very robust in conventional measuring environments like that in this study. The gyroscope performances of the 8-mm disk resonators are reported in Ref. [1-5] of the Supplementary Information, which can support the reliability of the experiment in this study.

Changes made:

- From Line 132 to Line 151 of the revised main document, we explain the rotation-induced Coriolis coupling.
- From Line 60 to Line 72 of Supplementary Information and in Supplementary Figure 3a, a detailed description of the read-out circuitry has been provided.
- From Line 81 to Line 89 of Supplementary Information and in Supplementary Figure 3c, the experimental setup used to rotate the micromechanical resonator has been explained.
- In Line 127 of the main document and Line 49 of Supplementary Information, we have made a statement that the deformations of the operational modes are perfectly in-plane.
- From Line 138 to Line 141 of the main document, we have made an explanation about the rate table.

Another attractive result of the manuscript is the demonstration of a physical system where the nonreciprocity can be controlled electrostatically. It would be helpful for the readers to include a discussion on how much the nonreciprocity can be augmented in linear systems using the new “knob.”

Response: We thank the Referee for this constructive suggestion. The nonreciprocal process in this study is very similar to the dynamical encircling of the EP singularity (e.g. Ref. [17,18] of the main document). Following a closed loop that circles the EP, the clockwise or counter-clockwise traversing would result in different final states. The clockwise encircling will irreversibly transfer the lower state to the higher state; The counter-clockwise encircling will irreversibly transfer the higher state to the lower state. In this study, we can obtain a similar effect, the difference is that we can observe such nonreciprocal state transfer by traversing a voltage-controlled 1D closed loop. The down-up-down trajectory will irreversibly transfer the higher state to the lower state, while the up-down-up trajectory will irreversibly transfer the lower state to the higher state.

The full state information of the starting and ending states is recorded. In the revised manuscript, we extrapolate the state amplitudes of the starting and ending states. The asymmetric transforming matrices that describe such nonreciprocal state transfer are obtained. The ratio of the nondiagonal elements of the asymmetric transforming matrix is used to characterize the isolation of the nonreciprocity. Based on the measured data, an isolation ratio of 59 dB (−43 dB) for the up-down-up (down-up-down) traversing

process is obtained.

Changes made:

- From Line 318 to Line 332 of the revised manuscript, we have made a background explanation of the singularity-related nonreciprocal state transfer.
- From line 345 to line 361, a more detailed description of the nonreciprocal state transfer is provided.
- In Supplementary Note 9, an extended explanation of the calculation of the transforming matrix and the isolation ratio is provided.

The manuscript claims that "although we experimentally demonstrate the phase tomographic closed-loop cubic singularity based on the Coriolis coupling, in principle, it can also be realized using ordinary linear coherent coupling." I don't understand this statement. If this is true, what is the advantage of the Coriolis force? A linear coherent coupling could be implemented using electronic circuitry among the resonators, which significantly simplifies the implementation of this idea.

Response: We apologize for this confusing statement. We have modified the statement in the revised manuscript. Our experimental demonstration in this paper focuses on Coriolis coupling, which makes this effect very useful for enhancing gyroscope performance. For example, using the high sensitivity of the singularity nexus, we can realize deep-sub-linewidth mode matching, which remains a remarkable challenge in the Coriolis vibratory gyroscope community. Besides, by modifying the configuration, we can realize the amplification of the Coriolis effect based on this cubic singularity, which will be reported in our follow-up work.

It is theoretically possible to realize the same effects using ordinary linear coherent coupling, but it still needs more verification. This is what we want to pursue in our future research.

Changes made:

- From Line 392 to Line 398 of the Discussion part of the main document, we have made a statement to prevent confusion. "The PhT cusp singularity resulting from Coriolis coupling can be directly used to enhance gyroscope sensitivity and achieve deep-sub-linewidth mode matching. While our experimental demonstration focuses on Coriolis coupling, it is theoretically possible to realize the same effects using ordinary linear coherent coupling (see Supplementary Discussion)."

In summary, the manuscript reports an interesting prototype to study singularities in coupled binary systems without involving nonlinearities, but it requires substantial modifications and does not seem a good fit for Nature Communications.

Response: Once again, we sincerely thank the Referee for giving such constructive comments and suggestions. We have carefully studied and responded to all the comments, and made considerable modifications to address those concerns, which we hope meet with approval.

Reviewer #3 (Remarks to the Author):

The paper describes an experimental observation of hysteretic phenomena in a system of two coupled micromechanical modes. The detuning of the modes and the coupling are the control parameters. One of the modes is driven close to resonance. The central (albeit unannounced) idea is that the system can be made effectively nonlinear and that this is accomplished by fixing the phase difference between the vibrations of the driven mode and the drive. Unexpectedly, this nonlinearity leads to the multi-branch response and, respectively, to hysteresis. The qualitative features of the emerging behaviour are familiar from the bifurcation theory, but this theory had not been applied to the system studied in the paper. The observations are new. I find them interesting.

Response: We sincerely thank the Referee for the positive assessment of our work and the excellent comments. The revision guided by those comments has greatly enhanced our paper.

We thank the Referee for reminding us that the idea of using phase tracking to realize cubic bifurcation is unannounced in the paper. As the Referee suggested, we have made a statement that our study provides an alternative approach for building cubic bifurcation by constructing a phase-tracked closed-loop oscillation of a coupled linear system without the need for nonlinear vibration.

Changes made:

- In Line 380 of the revised Discussion Section, we have made a statement that “Furthermore, we present an alternative approach for creating cubic bifurcations by establishing a phase-tracked closed-loop oscillation in a coupled system without relying on nonlinearity. This not only enhances our understanding of closed-loop oscillation dynamics but also extends coherent control into the singularity region.”

Some details: The coupled modes are standing waves formed by whispering gallery modes in a micromechanical system that has the shape of a disk. The circular symmetry of the system makes the modes almost degenerate. Rotation of the disk leads to mode coupling via the Coriolis force. The coupling is controlled by the angular frequency. One of the modes is driven, its phase is measured, and the displacements of the modes are detected. The phase difference between the vibrations of the driven mode and the drive is set to $\pi/2$.

The frequency of the drive that maintains the phase difference is a function of the mode coupling and the frequency detuning. It satisfies a cubic equation. If plotted as a surface with respect to the control parameters, it has two folds that merge at the nexus. The folds are the sets of the bifurcation values of the control parameters.

A few comments now.

I think the paper has to be rewritten. It has real results, there is no need to oversell them. Terms like “phase-tomographic singularity” are irrelevant and confusing.

Response: We thank the Referee for this constructive suggestion. As the Referee

suggested, we have rewritten this manuscript following all the feedback received. It is really true that the definition of “phase-tomographic singularity” might lead to confusion. We have also considered the definition of “phase-locked singularity”, but the expression of “phase-locking” has an existing meaning that is related to the synchronization effect and is totally different from the topic of this study, so it would lead to more serious confusion. The essence of the control in this study is to track the driven-mode phase response. Therefore, we define it as “phase-tracked (PhT) singularity” in the revised manuscript.

Changes made:

- We have rewritten the manuscript based on all the feedback received.
- We have replaced the “phase-tomographic singularity” with “phase-tracked (PhT) singularity” throughout the paper.

The ‘stability’ has to be defined. In the text, there is a reference to the stability of the drive frequency, but I believe that the stability refers to the mode vibrations. The corresponding equations have to be provided and loss of stability when crossing the folds has to be demonstrated.

The picture of the drive frequency as a function of the parameters is a standard picture of the bifurcation theory, see Ref. 1. It is drawn in multiple textbooks and monographs (a recent one is Kamenev’s “Field Theory of Non-Equilibrium Systems”). There are standard names for the corresponding bifurcations, co-dimension 1 for the folds (usually the saddle-node bifurcation) and co-dimension 2 for the cusp. These terms have been broadly used in micro- and nanomechanics, for example in the analysis of the bistability of the response of nonlinear modes to a resonant drive. The analysis can be and should be formulated in standard terms.

Response: We thank the Referee for pointing out this important issue. The stability of the phase-tracked states is indeed the vibrational stability. We have included the stability analysis following the standard bifurcation theory in Supplementary Note 4 of the revised Supplementary Information. The phase-tracked states are the fixed points of the phase-tracked dynamics. The stability of each state is determined by the Jacobian matrix of the phase-tracked dynamical equations, which describes the divergence or convergence of the system near the fixed points with small perturbations. If the real parts of all the Jacobian eigenvalues are negative, the perturbed system near the fixed point is convergent. In this case, the phase-tracked state is stable. Otherwise, if any of the Jacobian eigenvalues have positive or zero real parts, the phase-tracked state will be unstable or critically stable, respectively. The critically stable phase-tracked states are referred to as the singularities.

In the revised Supplementary Figure 8, the process of losing stability when crossing the singularities has been demonstrated by showing all the eigenvalue real parts of the Jacobian matrix.

Changes made:

- In the revised Supplementary Information, we have included the stability analysis in Supplementary Note 4. The corresponding equations have been provided.
- In Supplementary Figure 8, we show the calculated real Jacobian eigenvalues of some typical cross sections. The loss of stability when crossing the singularities has been demonstrated.
- The description of the stability of the phase-tracked state has been revised accordingly in Line 108 of the main document.
- We have included the book “Field Theory of Non-Equilibrium Systems” in our reference list.

To continue on the terminology: There are no phase transitions in the studied system and there are no phases of matter, there are phases of the modes. Calling hysteresis “nonreciprocity” does not add to the understanding, just makes it confusing to see what is being discussed. Calling the phase shift between the modes “circular polarization” does not help to understand what is going on. Panel (d) in Fig. 2 makes an impression that the “circular polarization phase” emerges immediately after crossing the bifurcation point, which is incorrect, I guess. Besides, the term “phase” here refers, judging from the context to the “phase of matter”, although this is, in fact, a vibrational phase.

Response: We thank the Referee for reminding us of the terminology in this paper. There is indeed no transition of matter phase. The ‘phase’ used in this paper indicates the vibrating state. Recently, in analogy but differently from the real matter phase transition, the transitions of different vibrating or oscillating states in photonic or acoustic resonators are also regarded as a generalized phase transition. A typical example is the parity-time-symmetric phase and the parity-time-symmetry broken phase [e.g. Miri et al., Science 363, eaar7709 (2019)]. Sometimes, the nonlinear oscillating state transition is also regarded as a generalized phase transition [e.g. Arkadev et al., Nature Physics, 19, pages 427–434 (2023)]. This is because those oscillating states can be classified by the purities of some vibrating characteristics, and the effective order parameter can be defined to describe the population/purity of different characteristics in oscillating states very similar to the definition of the order parameter of the matter phase. As the Referee suggested, we avoid the use of “phase transition” in the revised manuscript. Instead, we call it “the transition of oscillation phase”.

We would like to thank the Referee for pointing out the error in describing the oscillation phase above the singularity nexus as the “circular polarization phase” in Fig. 2d of the original manuscript. It should be called the “chiral symmetry broken phase” because the linear polarization evolves to the ellipse polarization at first. In the revised manuscript, this error has been corrected. This representation mistake did not appear in the main text.

We apologize for the unclear description of the nonreciprocal state transfer in the original manuscript. It was previously demonstrated that by dynamically encircling the

second-order EP singularity, one can observe a nonreciprocal state transfer [e.g. H. Xu, et al, Nature 537, 80 (2016)]. Recently, it has been proved that the nonreciprocal state transfer can also be realized by encircling the cubic EP singularity [e.g. H. Wang, et al., Optics Letters 44, 638 (2019)]. Moreover, it is also discovered that the winding process is not a must for this nonreciprocity. By taking advantage of the non-trivial landscape of the spectrum surface near the singularity, nonreciprocal state transfer can be realized by traversing an EP-excluding cycle [e.g. H. Nasari, et al., Nature 605, 256 (2022)]. Here, we show that the PhT cubic singularity can also produce a closed-loop controlled nonreciprocal state transfer. Besides, compared to the previous studies that depend on two-parameter-controlled encircling, we can realize the nonreciprocal state transfer by traversing a single-parameter-controlled 1D closed loop. The down-up-down trajectory will irreversibly transfer the higher state to the lower state, while the up-down-up trajectory will irreversibly transfer the lower state to the higher state. By extrapolating the amplitudes of the starting and ending states, we can describe such nonreciprocal state transfer using an asymmetric transforming matrix. The ratio of the nondiagonal elements of the asymmetric transforming matrix is used to characterize the isolation of the nonreciprocity. Based on the measured data, an isolation ratio of 59 dB (−43 dB) for the up-down-up (down-up-down) nonreciprocal process is obtained. Benefiting from the electrostatic tunability of our device, the PhT cubic singularity has provided a desirable voltage-controlled nonreciprocity. In the revised paper, we have included a more detailed explanation of the background and the details of the nonreciprocal state transfer process.

Changes made:

- We have replaced the representation of “phase transition” with “transition of oscillation phase” throughout the revised manuscript.
- The “circular polarization phase” representation in the original Figure 2d has been corrected.
- From Line 318 to Line 332 of the revised manuscript, we have made a background explanation of the singularity-related nonreciprocal state transfer.
- From line 345 to line 361 of the revised manuscript, a more detailed description of the nonreciprocal state transfer is provided.
- In Supplementary Note 9, an extended explanation of the calculation of the transforming matrix and the isolation ratio is provided.

I believe the mapping on the Bloch sphere is counterproductive. It makes the formulation look “quantum”, but rather distracts from what is going on. The modes are dissipative, and there is no Poisson bracket between the variables. Projecting the folds on the Bloch sphere instead of the standard projection on a plane makes the picture confusing.

Response: We apologize for the improper organization of the original manuscript, which made the state information described by the classical Bloch sphere hard to

understand. We have reorganized the manuscript to solve this problem. In the revised Results Section of the paper, we first introduce the concept of the PhT singularity. Then, we describe the experimental realization and observation of the PhT singularity. The state information behind the PhT frequency is separately explained in the following subsection, which would not distract from understanding the catastrophe effect on the PhT frequency surface.

We agree with the Referee that this singularity should be described classically. In the revised Supplementary Notes 2 and 3, we have replaced the original quantum description with the classical theory. The operators are replaced with variables, the Hermitian conjugate ‘†’ is replaced with the complex conjugate ‘*’, and the quantum Poisson bracket or the commutator ‘[]’ is replaced with the classical Poisson bracket ‘{}’.

In our opinion, state information is very important and even indispensable for explaining the singularities for two reasons. First, just like the eigenstates that are described by eigenfrequencies and the corresponding eigenvectors, the PhT states are also described by PhT frequencies and the corresponding state vectors. The PhT frequency is an index, but the PhT state vector is the essence. Second, demonstrating the PhT state vector is crucial to understanding the physical mechanism of the PhT singularity. By studying the state information behind each PhT frequency, we show that the “pitchfork” bifurcation is caused by the breaking of chiral symmetry, and the singularities are associated with transitions of oscillation phases with different chirality. Only by studying the PhT state vector, we can figure out the evolution of the oscillation phases.

One of the best ways to describe a state vector is to use the “bra-ket” notation because it can concisely and unambiguously describe an abstract vector without the need to predefine a basis. The projection of different states can be easily described by the “bra-ket” inner product. Though the states in this study are all classical, the “bra-ket” notation can still fully describe them.

To illustrate the state information of this two-mode system with a single driving frequency, we employ the classical Bloch sphere, which is equivalent to the Poincaré sphere characterized by the Stokes parameters. The difference is the choice of gauge. In the classical Bloch sphere that we use, the north (south) pole is the standing-wave eigenstate $|1\rangle$ ($|2\rangle$) that can be directly measured in our setup. For the Poincaré sphere, the north (south) pole is the $|CW\rangle$ ($|CCW\rangle$) chiral state, which is a travelling-wave state that cannot be directly measured in our setup. Since the state vectors are more convenient to be described in standing-wave eigenstate basis $\{|1\rangle, |2\rangle\}$, we use the classical Bloch sphere in this study.

Changes made:

- We have reorganized the Results Section of the manuscript. First, we introduce the concept of the phase-tracked singularity. Next, we describe the experimental realization and observation of the PhT singularity. Then, we explain the state

information behind the PhT frequency. We show that the “pitchfork” bifurcation is caused by the breaking of chiral symmetry, and the singularities are associated with transitions of oscillation phases with different chirality.

- Figure 2 of the original main document is separated. The revised Figure 2 describes the experimental realization of the PhT singularity. The revised Figure 3 explains the state information.
- In the revised Supplementary Notes 2 and 3 and Supplementary Discussion, we have replaced the original quantum description with the classical theory. The operators are replaced with variables, the Hermitian conjugate ‘†’ is replaced with the complex conjugate ‘*’, and the quantum commutator ‘[]’ is replaced with the classical Poisson bracket ‘{}’.
- From Line 213 to Line 280 of the revised main document, A new state information subsection is additionally provided in the revised manuscript. The corresponding representation is reorganized. Some of the major modifications include: From Line 213 to Line 217 of the revised manuscript, we made a statement that “In the following, we will delve into the details of the state information corresponding to each PhT frequency. We will show that the pitchfork bifurcation is caused by the breaking of chiral symmetry, and the singularities are associated with transitions of oscillation phases with different chirality.” In Line 222 of the revised manuscript, we made a statement that “It is important to note that all states involved in this study are classical.” From Line 247 to Line 250 of the revised manuscript, we explain that the frequency bifurcation is produced by the spontaneous breaking of chiral symmetry and the rotational Doppler effect. Some other modifications are also made to adapt to the context.
- We have redefined the order parameter as the relative population of the CW and CCW states, which equals the chirality. The sign of the order parameter is reversed compared to that of the original manuscript.

If the authors want to put emphasis on the phase shift between the modes and the change of their amplitude, they could use the Stokes parameters. This is unnecessary, but at least this is a standard language for a two-mode system where the modes vibrate at the same frequency.

Response: We thank the Referee for the suggestion of using Stokes parameters to describe the state information. As the Referee suggested, we have provided the Stokes parameters S_1 , S_2 , and S_3 in the main document. Here, the three parameters are the sphere coordinates of the classical Bloch sphere, which is the rotated Poincaré sphere.

Changes made:

- From Line 225 to Line 227 of the revised main document, the Stokes parameters are provided, where S_1 , S_2 , and S_3 stand for ellipticity, chirality, and orientation, respectively.
- From Line 180 to Line 183 of the revised Supplementary Information, the

explanations of the Stokes parameters are provided.

- The state information is provided in Figure 3a of the revised manuscript.

I note also that the high sensitivity of a mechanical system near the cusp point (nexus) has been demonstrated by Aldridge and Cleland [PRL 94, 156403 (2005)]; the authors demonstrated that, in fact, the sensitivity can be made exponentially strong if one looks at the response to fluctuations in the corresponding parameter range. However, the scaling of the dynamics is certainly the same as in the present paper.

Response: We thank the Referee for reminding us of this related work. It is a very important demonstration of sensitivity enhancement provided by the nonlinearity-related (cubic) cusp point. We have included this paper in reference.

Given the novelty of the results, a significantly revised version of the paper can be considered for publication in Nature Communications.

Response: Once again, we sincerely thank the Referee for the constructive comments. We have tried our best to address all these concerns and suggestions. The revisions based on the comments have strengthened our paper from the aspects of significance and stringency. We hope our revisions and responses meet the expectations of the Referee.

REVIEWER COMMENTS

Reviewer #2 (Remarks to the Author):

The latest version of the manuscript has improved significantly. Most of the issues raised by the reviewers have been addressed properly. The manuscript is now suitable for publication in Nature Communications.

Reviewer #3 (Remarks to the Author):

The authors have significantly modified the paper. However, more work is needed before I would feel comfortable recommending the paper for publication. I reiterate that, from my point of view, the observations are interesting.

In their rebuttal the authors agree with the comments and acknowledge the mistakes in the first version. They have redrawn the Bloch sphere using the Stokes parameters and made a few more important changes. However, in the text they still keep much of the old language. Moreover, wrong statements have been added.

It is my impression that the authors are trying to hide behind artificially introduced terms to make the paper look more significant. In fact, the paper would be more impressive if it were made understandable and put into the proper context.

Specific comments:

1. I again encourage the authors to look more closely at the standard text [1] or any other text on the bifurcation theory. The folds do not intersect, contrary to the statement in the paper, they merge at the cusp point. The “singularity arc” – is this just the projection of the fold on the parameter space, the standard line of saddle-node bifurcations? What is “cubic singularity arc”? What is “remarkable” about the standard structure of the bifurcation lines? Why not use standard terms?

2. It is true that, if you vary a certain combination of the parameters, the cusp point looks like a pitchfork bifurcation. No quotes are needed. However, there are no “unbalanced pitchfork bifurcations”. Is the

term used here to describe standard saddle-node bifurcations? What is the difference with the standard analysis of the dynamics near a cusp point?

Also, there are no “degenerate” bifurcations. There are no "cubic bifurcations".

3. The authors keep talking about “nonreciprocity” instead of the standard hysteresis. Hysteretic response is very familiar in nano and micromechanics. Sophisticated 3D color pictures do not add to what is already well understood. I think this part can be shortened or eliminated.

4. I believe the Introduction should be rewritten. It is put into the title and italicized in the text that the system is “binary linear”. In the rebuttal the authors agree that the system is strongly nonlinear as, in addition to the two modes, it contains a PLL, a strongly nonlinear control device. It is the nonlinearity that makes the system interesting.

The claim of the novelty of the onset of folds and cusp points in a two-mode system has to be removed. The corresponding bifurcations have been well-known even for a single nano/micromechanical mode and have been described in the experimental paper the authors are now referring to (Ref.37); more examples and more types of bifurcations observed in nanomechanics as well as the underlying theory can be found in Bachtold et al., Rev. Mod. Phys. (2022).

It is also well-known in nanomechanics that, because nanomechanical systems have high quality factors, there is no need in “overexcited conditions, which ... bring intrinsic power consumption and reliability limits”. Already weak resonant drives lead to a variety of strongly nonlinear effects, including dynamical chaos, cf. the above RMP and in particular Guttinger et al., Nat. Nano (2017), and Hourii et al., PRL (2020).

The interesting and unexpected result of the paper, from my point of view, is that, by attaching a PLL to a MEMS gyroscope, it is possible to make the driven system bistable in a way qualitatively different from the familiar bistability due to the Duffing nonlinearity and to investigate the emerging bifurcations. I would emphasize this.

Response to Reviewers

We express our sincere gratitude to the reviewers for their kind reviews. The manuscript has been revised to address the feedback received. Please find below our response to the reviewers. Reviewers' comments are in **black**, authors' responses are in **Blue**, and the **Red parts** indicate the corresponding modifications in the manuscript.

Reviewer #2 (Remarks to the Author):

The latest version of the manuscript has improved significantly. Most of the issues raised by the reviewers have been addressed properly. The manuscript is now suitable for publication in Nature Communications.

Response: We sincerely thank the Referee for the valuable comments, which have greatly improved our paper.

Reviewer #3 (Remarks to the Author):

The authors have significantly modified the paper. However, more work is needed before I would feel comfortable recommending the paper for publication. I reiterate that, from my point of view, the observations are interesting.

In their rebuttal the authors agree with the comments and acknowledge the mistakes in the first version. They have redrawn the Bloch sphere using the Stokes parameters and made a few more important changes. However, in the text they still keep much of the old language. Moreover, wrong statements have been added.

It is my impression that the authors are trying to hide behind artificially introduced terms to make the paper look more significant. In fact, the paper would be more impressive if it were made understandable and put into the proper context.

Response: We express our sincere gratitude to the Referee for the constructive comments and valuable suggestions. We are truly grateful to the Referee for providing us with the opportunity to present our results in a more unambiguous manner. Additionally, we also thank the Referee for bringing to our attention the nonstandard representations, which we have now revised based on the standard representations from refs. *V. I. Arnold, Catastrophe Theory (Springer-Verlag, 1984)* and *S. H. Strogatz, Nonlinear dynamics and chaos: with applications to physics, biology, chemistry, and engineering, 2nd ed. (CRC Press, 2015)*.

Specific comments:

1. I again encourage the authors to look more closely at the standard text [1] or any other text on the bifurcation theory. The folds do not intersect, contrary to the statement in the paper, they merge at the cusp point. The “singularity arc” – is this just the projection of the fold on the parameter space, the standard line of saddle-node bifurcations? What is “cubic singularity arc”? What is “remarkable” about the standard

structure of the bifurcation lines? Why not use standard terms?

Response: We thank the Referee for pointing out the nonstandard representations. In the revised manuscript, we have corrected the nonstandard statements such as “intersect”, “singularity arc”, and “cubic singularity arc”.

Changes made:

- From Line 53 to Line 62 of the revised main text, we correct the representations about the singularities: “Notably, by examining the equiphase contour of the coherent-coupling phase response, we find that the system can exhibit bistability in a way qualitatively different from the Duffing nonlinearity. The boundaries of stability are constituted by a series of **saddle-node bifurcation points**, leading to the singularity named **folds** to describe the abrupt transitions that occur during parametric sweeping across these boundaries. Two folds **tangentially merge** at a pitchfork bifurcation point referred to as a nexus, which defines a cusp singularity if projected onto the parameter plane.”
- From Line 113 to Line 120 of the revised main text, we correct the representations about the singularities: “If the control parameters g and $\Delta\omega$ steer across the stability boundaries made by the **saddle-node bifurcation points** adiabatically, catastrophic jumps in the oscillation state take place, defining the singularity called **folds**. The folds **tangentially merge** at the pitchfork bifurcation point (star in Fig. 1d), giving rise to a nexus and a markedly twisted ω_d^* geometry. The projection of the nexus onto the $\Delta\omega$ - g parameter plane defines a cusp singularity.”
- From Line 258 to Line 262 of the revised main text, we correct the representations about the singularities: “A series of **saddle-node bifurcation points** (white-faced points) on the lower hemisphere constitute the **folds** (blue curves). Two folds **tangentially merge** at the pitchfork bifurcation point referred to as the nexus (blue point), forming a cusp singularity.”
- In the revised caption of Fig.3a, we make a statement: “The blue curve represents the singularities, which are composed of a series of **bifurcation points**.”
- Some other related minor modifications in the main text and the Supplementary Information are also made for the cohesiveness of the content.

2. It is true that, if you vary a certain combination of the parameters, the cusp point looks like a pitchfork bifurcation. No quotes are needed. However, there are no “unbalanced pitchfork bifurcations”. Is the term used here to describe standard saddle-node bifurcations? What is the difference with the standard analysis of the dynamics near a cusp point?

Also, there are no “degenerate” bifurcations. There are no “cubic bifurcations”.

Response: We thank the Referee for pointing out the nonstandard representations such as “unbalanced pitchfork bifurcations”, “singularity arc”, “cubic singularity arc”, “degenerate bifurcation”, and “cubic bifurcations”. They have been revised using

standard representations.

Changes made:

- In Lines 99, 184, and 272 of the revised main text, the “degenerate bifurcation point” in the original manuscript is replaced by the “pitchfork bifurcation point”.
- From Line 102 to Line 103 of the revised main text, the statement about the bifurcation has been revised: “As the degeneracy is broken, the pitchfork bifurcation of ω_d^* disconnects into a stable branch and a saddle-node bifurcation.”
- From Line 109 to Line 111 of the revised main text, the statement about the bifurcation has been revised: “The inflectional region (highlighted in red) within the folded ω_d^* surface in Fig. 1d is made by the unstable bifurcation branches.”
- From Line 185 to Line 189 of the revised main text, the statement about the bifurcation has been revised: “In cases where the degeneracy is broken, the symmetry of normal mode splitting in the $|q_1|$ responses is broken, and the $\theta_1 = -\pi/2$ equiphase contour in the θ_1 responses illustrates a stable branch and a saddle-node bifurcation.”
- In Line 235 of the revised main text, the “balanced pitchfork bifurcation” in the original manuscript is replaced by the “pitchfork bifurcation”.
- From Line 253 to Line 258 of the revised main text, the statement about the state evolutions of the bifurcations has been revised: “In the degeneracy-broken case, the state evolutions associated with the bifurcation patterns in Fig. 2e and f are shown by the orange and magenta trajectories on the Bloch sphere in Fig. 3a, respectively. The introduction of rotation immediately leads to the breaking of chiral symmetry, as shown by the stable branches on the upper hemisphere.”
- From Line 331 to Line 334 of the revised main text, the statement about the bifurcations has been revised: “Furthermore, we present an alternative approach for creating bistability and bifurcations by establishing a phase-tracked closed-loop oscillation in a coupled system without relying on nonlinear potential energy.”
- In the revised caption of Fig.2d, e and f, the term “unbalanced pitchfork bifurcations” has been replaced by “saddle-node bifurcations”.
- In the revised caption of Fig.3b, the “degenerate bifurcation” has been replaced by “pitchfork bifurcation”.
- Some other related minor modifications in the main text and the Supplementary Information are also made for the cohesiveness of the content.

3. The authors keep talking about “nonreciprocity” instead of the standard hysteresis. Hysteretic response is very familiar in nano and micromechanics. Sophisticated 3D color pictures do not add to what is already well understood. I think this part can be shortened or eliminated.

Response: We sincerely appreciate the Referee for reminding us of the similarities

between the nonreciprocal traversal process in this study and the hysteresis sweep process in Duffing nonlinear frequency responses. We accept the suggestion of the Referee to shorten the section in the main text about the nonreciprocal state transfer.

However, we also note that the PhT singularity explored in our research differs slightly from the Duffing nonlinearity-induced bifurcation, particularly in terms of the underlying state information for each bistability. In the case of Duffing nonlinearity-induced bifurcation, the bistable states are associated with the same mode, without any mode transfer occurring when the states are switched. In contrast, for the hysteretic effect in our study, the bistable states correspond to two distinct linear oscillating modes located at different regions on the Bloch sphere. Switching between these bistable states enables state transfer, making our PhT singularity more akin to exceptional point singularities.

Given that the nonreciprocal state transfer is not the primary innovation of this paper, we have followed the Referee's suggestion. We have relocated most of the content regarding nonreciprocal state transfer to SUPPLEMENTARY DISCUSSION 1. In the revised DISCUSSION section of the main text, we provide a concise statement about nonreciprocal state transfer, highlighting it as one of the supporting applications of the PhT singularity.

Changes made:

- The “Voltage-controlled nonreciprocity” section and Fig. 5 of the original manuscript have moved into the SUPPLEMENTARY DISCUSSION 1 of the revised Supplementary information.
- From Line 342 to Line 349 of the revised main text, we make a succinct statement about the voltage-controlled nonreciprocity state transfer: “Additionally, the PhT cubic singularity can also facilitate the realization of closed-loop controlled nonreciprocal state transfer (see Supplementary Discussion 1). In contrast to previous studies that rely on two-parameter-controlled encircling, we demonstrate the achievement of nonreciprocal state transfer through the highly desirable single-parameter (voltage)-controlled traversal, resulting in an impressive isolation ratio of 59 decibels.”

4. I believe the Introduction should be rewritten. It is put into the title and italicized in the text that the system is “binary linear”. In the rebuttal the authors agree that the system is strongly nonlinear as, in addition to the two modes, it contains a PLL, a strongly nonlinear control device. It is the nonlinearity that makes the system interesting.

The claim of the novelty of the onset of folds and cusp points in a two-mode system has to be removed. The corresponding bifurcations have been well-known even for a single nano/micromechanical mode and have been described in the experimental paper the authors are now referring to (Ref.37); more examples and more types of bifurcations observed in nanomechanics as well as the underlying theory can be found in Bachtold et al., Rev. Mod. Phys. (2022).

It is also well-known in nanomechanics that, because nanomechanical systems have high quality factors, there is no need in “overexcited conditions, which ... bring intrinsic power consumption and reliability limits”. Already weak resonant drives lead to a variety of strongly nonlinear effects, including dynamical chaos, cf. the above RMP and in particular Guttinger et al., Nat. Nano (2017), and Hourii et al., PRL (2020).

The interesting and unexpected result of the paper, from my point of view, is that, by attaching a PLL to a MEMS gyroscope, it is possible to make the driven system bistable in a way qualitatively different from the familiar bistability due to the Duffing nonlinearity and to investigate the emerging bifurcations. I would emphasize this.

Response: As the Referee suggested, we have rewritten the Introduction and rephrased the contribution of this study as “We demonstrate theoretically and experimentally the existence of an unexplored third-order singularity in the phase-tracked steady states of a pair of coherently coupled mechanical modes. Notably, by examining the equiphase contour of the coherent-coupling phase response, we find that the system can exhibit bistability in a way qualitatively different from the Duffing nonlinearity.” The title of the manuscript is also revised as “Higher-order singularities in phase-tracked electromechanical oscillators”.

The bistability of this study indeed can be regarded as an effective nonlinearity because the governing equation is cubic. Nonetheless, we also concur with the Referee in noting that the PhT singularity mechanism is fundamentally distinct from the Duffing nonlinearity-induced bifurcation. In the case of the PhT singularity, the bifurcation pattern is solely determined by the landscape of the coherent-coupling phase response, as illustrated in Figure 1b (also provided below). The role of the PLL is merely to track the equiphase contour (depicted as the black curve). Consequently, we believe that the nature of the PhT singularity is primarily predicated upon the linear response of the coherently coupled modes. The PLL serves as a technique to track this contour. Hence, we feel that it would be irresponsible for us to attribute the PhT singularity solely to the nonlinear control of the PLL. While some other schemes might lead to more intriguing linear or nonlinear phase-tracked patterns, the nature of these patterns would still be governed by the landscapes of the linear phase responses, rather than the nonlinear control of the PLL. Nevertheless, we acknowledge that the nonlinear control device PLL is crucial for executing the phase tracking. Following the Referee's suggestion, we have removed the claim regarding the singularity being linear in the revised manuscript. We think focusing on the actual results would be better.

We also would like to thank the Referee for reminding us that the Duffing nonlinearity can introduce pitchfork bifurcations and cusp singularities in a single mode. Based on this, we have revised the Introduction part and made a statement that “**Interestingly, nonlinearities can facilitate the emergence of higher-order singularities, such as dynamical pitchfork bifurcation points and higher-order exceptional points, while requiring fewer degrees of freedom.**” The related references including Bachtold et al., Rev. Mod. Phys. (2022) have been included. we have also removed the emphasis on the binary nature of the PhT singularity in the revised manuscript following the Referee’s suggestion.

Generally, conventional nonlinearity relies on nonlinear potential energies that can provide nonlinear restoring forces. We wholeheartedly agree that nanomechanical resonators can readily achieve nonlinearity. As advised by the Referee, we have removed the improper claim from the original paper that mentioned “overexcited conditions, which ... bring intrinsic power consumption and reliability limits.”

However, in micro/mesoscale mechanical resonators, we must acknowledge that nonlinearity is still relatively less controllable compared to established techniques such as PLL and coherent coupling, which have been extensively utilized in the development of mature devices.

We deeply appreciate the Referee's overall perspective regarding the contribution of this study. We agree with the Referee's viewpoint that through PLL-enabled phase tracking, we can achieve a qualitatively distinct bistability in the driven coupled system compared to the familiar bistability resulting from Duffing nonlinearity. This enables us to investigate emerging bifurcations in a novel manner. As advised by the Referee, we have revised the Abstract, Introduction, and Discussion sections to reflect these changes.

Changes made:

- From Line 35 to Line 43 of the revised main text, we have rewritten the background of this study: “**Interestingly, nonlinearities can facilitate the emergence of higher-order singularities, such as dynamical pitchfork bifurcation points and higher-order exceptional points, while requiring fewer degrees of freedom. Exploring these phenomena not only expands our understanding of singularity dynamics but also**

paves the way for engineering controllable devices. Nevertheless, these nonlinearities are often associated with well-established nonlinear potential energies.”

- From Line 50 to Line 56 of the revised main text, we have rephrased the contribution of this study: “We demonstrate theoretically and experimentally the existence of an unexplored third-order singularity in the phase-tracked steady states of a pair of coherently coupled mechanical modes. Notably, by examining the equiphase contour of the coherent-coupling phase response, we find that the system can exhibit bistability in a way qualitatively different from the Duffing nonlinearity.”
- From Line 323 to Line 325 of the revised main text, we have rephrased the contribution of this study: “In summary, our study has discovered a cubic singularity in the phase-tracked coherent-coupling dynamics of a pair of microelectromechanical modes.”
- Some other related minor modifications in the main text and the Supplementary Information are also made for the cohesiveness of the content.

REVIEWERS' COMMENTS

Reviewer #3 (Remarks to the Author):

The paper has been significantly improved. As suggested in the report, the title has been changed and the emphasis on linearity has been removed. However, there are a few things that need to be cleaned up before I would recommend publishing it.

1. Singularities are not a phenomenon. Consequences of bifurcations are observable and could be called phenomena.

2. "Cubic singularities" – this is an obscure term. The authors are talking about a codimension-2 bifurcation, a cusp point on the parameter plane. And, in contrast to the claim in the paper, in the studied case reaching this point also requires "meticulous tuning", as in the case studied in Ref. 32 and in many other papers on nano- and micromechanical systems. In terms of applications, besides Ref. 32, this point was suggested as a means of suppressing phase noise in the very well-known paper by Greywall et al., PRL 72 (1994). This reference could have been found in the review [33] to which I referred earlier.

3. "Superior performance" has to be compared with ref. 32 and with the Greywall paper and subsequent papers by Kenig et al (PRE 86, (2012), PRE 88 (2013)).

4. A lot of experimental work on hysteresis and dynamics near bifurcation points in micro- and nanomechanical systems has been done over the years, in particular by the groups of Roukes, Ho Bun Chan, Collin, Kenny, Sheer, and others. I don't think this work should be disregarded.

5. A pitchfork bifurcation does not "disconnect into a stable branch...".

6. The last paragraph in red has, again, to be modified to explain the nature of the specific codimension-2 bifurcation studied in this paper.

Response to Reviewer

We express our sincere gratitude to the Referee for the kind reviews. The manuscript has been revised to address the feedback received. Please find below our response to the reviewers. The Referee's comments are in **black**, authors' responses are in **Blue**, and the **Red parts** indicate the corresponding modifications in the manuscript.

Reviewer #3 (Remarks to the Author):

The paper has been significantly improved. As suggested in the report, the title has been changed and the emphasis on linearity has been removed. However, there are a few things that need to be cleaned up before I would recommend publishing it.

Response: We extend our sincere thanks to the Referee for the valuable and constructive comments. The feedback has greatly enhanced our manuscript, both in terms of scientific content and representation.

1. Singularities are not a phenomenon. Consequences of bifurcations are observable and could be called phenomena.

Response: As the Referee suggested, the improper representation of "Singularities are a ubiquitous phenomenon..." has been revised in the revised paper.

Changes made:

- In the first sentence of the revised abstract, we correct the representations about the singularity: "Singularities ubiquitously exist in different fields and play a pivotal role in probing the fundamental laws of physics and developing highly sensitive sensors."

2. "Cubic singularities" – this is an obscure term. The authors are talking about a codimension-2 bifurcation, a cusp point on the parameter plane. And, in contrast to the claim in the paper, in the studied case reaching this point also requires "meticulous tuning", as in the case studied in Ref. 32 and in many other papers on nano- and micromechanical systems. In terms of applications, besides Ref. 32, this point was suggested as a means of suppressing phase noise in the very well-known paper by Greywall et al., PRL 72 (1994). This reference could have been found in the review [33] to which I referred earlier.

Response: We sincerely thank the Referee for the insightful comments regarding the representation of "Cubic singularities." It is true that the PhT singularity is classified as the codimension-two cusp singularity/catastrophe. As per the Referee's suggestion, we have revised the term in the manuscript to "cusp singularity." The codimension-two nature of the PhT singularity is also demonstrated in the Methods section.

In the abstract, the expression "meticulous" is to describe the tuning of multiple (≥ 3) coupled degrees, where at least six controlling parameters are involved. The tuning of the cusp singularities is much simpler because only two controlling parameters are involved. To avoid misunderstanding, the representation in the abstract has been revised.

We are grateful to the Referee for reminding us about the application of suppressing noise using the Duffing-nonlinearity-induced bifurcation point. As per the recommendation, we have included the recommended references in the revised manuscript.

Changes made:

- From Line 113 to Line 115 of the revised main text, we have made a statement that equation (1) “describes a cusp singularity because equ. (1) is right-equivalent to the universal unfolding of Thom's codimension-two catastrophe.”
- In the revised abstract, the statement has been revised as “Nevertheless, achieving higher-order (≥ 3) singularities, which exhibit superior performance, typically necessitates meticulous tuning of multiple (≥ 3) coupled degrees of freedom or **additional introduction of nonlinear potential energies**” to avoid misunderstanding.
- In Line 32 of the revised main text, “suppressing noise” has been added as one of the applications of singularities. The references recommended by the Referee are included as ref. [13-17].

3. “Superior performance” has to be compared with ref. 32 and with the Greywall paper and subsequent papers by Kenig et al (PRE 86, (2012), PRE 88 (2013)).

Response: As the Referee suggested, the recommended references about noise suppressing are included in the introduction “Higher-order singularities have the potential to provide higher performance and engender richer physics.”

Changes made:

- The recommended references are included as ref. [13-17] of the revised manuscript.

4. A lot of experimental work on hysteresis and dynamics near bifurcation points in micro- and nanomechanical systems has been done over the years, in particular by the groups of Roukes, Ho Bun Chan, Collin, Kenny, Scheer, and others. I don't think this work should be disregarded.

Response: As the Referee suggested, the recommended experimental studies about Duffing-induced bifurcation points are included in the introduction of the revised manuscript.

Changes made:

- The recommended references are included as ref. [15,17,38-43] of the revised manuscript.

5. A pitchfork bifurcation does not “disconnect into a stable branch...”.

Response: We apologize for the unclear representation, which has been revised in the latest manuscript.

Changes made:

- From Line 105 to Line 107 of the revised main text, we revise the description as “As the degeneracy is broken, the perturbed pitchfork bifurcation of ω_d^* splits into a saddle-node bifurcation and a stable branch.”

6. The last paragraph in red has, again, to be modified to explain the nature of the specific codimension-2 bifurcation studied in this paper.

Response: As the Referee suggested, in the Methods section, we have proved the codimension-two nature of the PhT singularity by demonstrating that the governing cubic equation (1) is right-equivalent to the universal unfolding of Thom’s codimension-two cusp catastrophe. By demonstrating this equivalence, we can classify the PhT singularity as a codimension-two cusp singularity.

Changes made:

- From Line 113 to Line 115 of the revised main text, we have made a statement that equation (1) “describes a cusp singularity because equ. (1) is right-equivalent to the universal unfolding of Thom's codimension-two catastrophe.”
- In the Methods section, we have additionally included a subsection “Codimension-two nature of the PhT singularity”.

Once again, we extend our gratitude to the Referee for the valuable input and suggestions, which have significantly improved the quality of our work.